# Diverse Methylmercury (MeHg) Producers and Degraders Inhabit Acid Mine Drainage Sediments, but Few Taxa Correlate with MeHg Accumulation

Jin Zheng,[a] Jie-Liang Liang,[a] Pu Jia,[a] Shi-wei Feng,[a] Jing-li Lu,[a] Zhen-hao Luo,[b] Hong-xia Ai,[b] Bin Liao,[b] (ID) Jin-tian Li,[a] Wen-sheng Shu[a]

[a]Institute of Ecological Science, Guangzhou Key Laboratory of Subtropical Biodiversity and Biomonitoring, Guangdong Provincial Key Laboratory of Biotechnology for Plant Development, School of Life Sciences, South China Normal University, Guangzhou, People's Republic of China
[b]School of Life Sciences, Sun Yat-sen University, Guangzhou, People's Republic of China

Jin Zheng and Jie-Liang Liang contributed equally to this work. Author order was determined in order of increasing seniority.

**ABSTRACT** Methylmercury (MeHg) is a notorious neurotoxin, and its production and degradation in the environment are mainly driven by microorganisms. A variety of microbial MeHg producers carrying the gene pair *hgcAB* and degraders carrying the *merB* gene have been separately reported in recent studies. However, surprisingly little attention has been paid to the simultaneous investigation of the diversities of microbial MeHg producers and degraders in a given habitat, and no studies have been performed to explore to what extent these two contrasting microbial groups correlate with MeHg accumulation in the habitat of interest. Here, we collected 86 acid mine drainage (AMD) sediments from an area spanning approximately 500,000 km$^2$ in southern China and profiled the sediment-borne putative MeHg producers and degraders using genome-resolved metagenomics. 46 metagenome-assembled genomes (MAGs) containing *hgcAB* and 93 MAGs containing *merB* were obtained, including those from various taxa without previously known MeHg-metabolizing microorganisms. These diverse MeHg-metabolizing MAGs were formed largely via multiple independent horizontal gene transfer (HGT) events. The putative MeHg producers from *Deltaproteobacteria* and *Firmicutes* as well as MeHg degraders from *Acidithiobacillia* were closely correlated with MeHg accumulation in the sediments. Furthermore, these three taxa, in combination with two abiotic factors, explained over 60% of the variance in MeHg accumulation. Most of the members of these taxa were characterized by their metabolic potential for nitrogen fixation and copper tolerance. Overall, these findings improve our understanding of the ecology of MeHg-metabolizing microorganisms and likely have implications for the development of management strategies for the reduction of MeHg accumulation in the AMD sediments.

**IMPORTANCE** Microorganisms are the main drivers of MeHg production and degradation in the environment. However, little attention has been paid to the simultaneous investigation of the diversities of microbial MeHg producers and degraders in a given habitat. We used genome-resolved metagenomics to reveal the vast phylogenetic and metabolic diversities of putative MeHg producers and degraders in AMD sediments. Our results show that the diversity of MeHg-metabolizing microorganisms (particularly MeHg degraders) in AMD sediments is much higher than was previously recognized. Via multiple linear regression analysis, we identified both microbial and abiotic factors affecting MeHg accumulation in AMD sediments. Despite their great diversity, only a few taxa of MeHg-metabolizing microorganisms were closely correlated with MeHg accumulation. This work underscores the importance of using genome-resolved metagenomics to survey MeHg-metabolizing microorganisms and provides a framework for the illumination of the microbial basis of MeHg accumulation via the

**Ad Hoc Peer Reviewer** (ID) Giovanni Gallo, Università degli Studi di Napoli Federico II

Address correspondence to Jin-tian Li, lijintian@m.scnu.edu.cn, or Wen-sheng Shu, shuwensheng@m.scnu.edu.cn.

The authors declare no conflict of interest.

characterization of physicochemical properties, MeHg-metabolizing microorganisms, and the correlations between them.

**KEYWORDS** acid mine drainage (AMD), demethylator, metagenome, methylator, methylmercury, sediment

Mercury (Hg) is a highly toxic contaminant that is released into the Earth's ecosystems by various mining and industrial activities (1, 2). The ecological and toxicological effects of Hg are strongly dependent on its chemical speciation, among which methylmercury (MeHg) is of special concern (3). As a neurotoxin, MeHg can bioaccumulate along the food chain, particularly in aquatic ecosystems, thereby intrinsically posing much higher risks to the health of humans and wildlife than Hg (4). Thus, understanding the factors controlling MeHg accumulation in aquatic environments is of high priority in order to reduce the health risks of Hg and its derivatives.

For most aquatic environments, MeHg is enriched mainly in sediments (5). Total Hg (THg) and MeHg concentrations in sediments can be up to five to seven orders of magnitude higher than those in their overlying waters, respectively (6, 7). Acid mine drainage (AMD) represents a type of widespread aquatic environment that is generated when solid mining wastes containing pyrite and/or other sulfide minerals are dissolved by microorganisms under conditions with oxygen and water (8). The low pH of AMD can accelerate the release and transfer of Hg from those minerals in which it is fixed. Generally speaking, the levels of THg and MeHg in AMD sediments are dramatically higher than those in the sediments of lakes, rivers, and oceans (9, 10). He et al. (11) showed that the concentrations of THg and MeHg in AMD sediments can be greater than 100 mg kg$^{-1}$ and 100 $\mu$g kg$^{-1}$, respectively. Therefore, AMD sediments have long been considered a hot spot for the study of MeHg accumulation (1). Moreover, AMD is considered to be a model system for microbial community studies due to its relatively simple microbial composition and its tight biological-geochemical coupling (12). These attributes also make AMD a good model for studying microbial involvement in MeHg accumulation. However, the diversity of MeHg-metabolizing microorganisms inhabiting AMD sediments and their effects on MeHg accumulation still remain poorly understood, although microbial processes are widely considered to be the dominant pathways of MeHg production and degradation in various sediments (13, 14).

Microbial MeHg producers, which are able to convert Hg(II) or Hg$^0$ into MeHg, generally carry a two-gene cluster, *hgcA* and *hgcB* (15). The *hgcA* gene encodes a corrinoid protein that serves as a methyl carrier, and the *hgcB* gene encodes a 2[4Fe-4S] ferredoxin that functions as an electron donor involved in corrinoid cofactor reduction (15). Due to the difficulty in obtaining pure cultures of MeHg producers, only 31 strains have been experimentally validated to have methylation activity (15–17). In recent years, genome-resolved metagenomic methods have become increasingly used to explore MeHg producers in different environments, such as oceans, fresh waters, and hot springs (18–20). Phylogenic analyses of nearly 1,000 metagenome-assembled genomes (MAGs) containing *hgcAB* from various environments revealed that these putative MeHg producers were distributed in more than 30 bacterial and archaeal phyla (19). Among well-known MeHg producers, a large proportion belong to sulfate-reducing bacteria (SRB), and the remaining are comprised mainly of iron-reducing bacteria (IRB), methanogens, and fermentative microorganisms (21).

The microbial demethylation of MeHg is the reverse process of Hg methylation, which can simultaneously occur in the locations where methylation takes place (5). This process can be performed by microorganisms via either a *merB*-mediated pathway or non-*merB*-mediated pathways (22). The *merB*-mediated pathway includes two main steps: microorganisms first cleave MeHg into Hg(II) and CH$_4$ with organomercury lyase (MerB), and then they further reduce Hg(II) to Hg$^0$ through mercuric reductase (MerA) (23). MerB and MerA are encoded by *merB* and *merA* of the *mer* operon, respectively. To date, the function of *merB* has been validated experimentally in only five bacterial

strains affiliated with two phyla (i.e., *Firmicutes* and *Proteobacteria*) (24, 25), although the currently available *merB* genes in the Pfam database (release 34.0) are distributed across 11 bacterial phyla and one archaeal phylum. Note also that only one *merB*-containing MAG affiliated with *Proteobacteria* has been reported in the literature (26). A non-*merB*-mediated (often referred to as "oxidative") pathway was observed in two bacterial strains (27, 28). In addition, a novel non-*merB*-mediated pathway of MeHg degradation has recently been revealed by Lu et al. (29) who demonstrated that a methanotroph was able to degrade MeHg, likely via methanol dehydrogenase encoded by *xoxF and mxaF*. Nonetheless, the elaborate genetic mechanisms underlying these two non-*merB*-mediated pathways still remain unclear.

In addition to MeHg-metabolizing microorganisms, certain abiotic factors (such as redox potential) are also thought to have profound effects on MeHg accumulation in the environment (21). In fact, the potential relationships between environmental factors, microbial functional genes (i.e., *hgcA*, *hgcB*, and *merB*) and MeHg levels in a variety of aquatic habitats other than AMD sediments have previously been explored (18, 20, 30, 31). However, the combined effects of microbial and abiotic factors on MeHg accumulation in the environment still remain poorly understood. It is similarly unclear how microbial MeHg producers and degraders can collectively affect MeHg accumulation in the environment, as evidenced by the fact that these two contrasting microbial groups were rarely simultaneously characterized in previous studies (20, 26).

In this study, we used genome-resolved metagenomics to explore the diversity, distribution, and abundance of putative MeHg producers and degraders in 86 AMD sediments collected from an area covering approximately 500,000 km$^2$ in southern China. A total of 46 MAGs carrying *hgcAB* and 93 MAGs carrying *merB* were obtained. They spanned seven and eight phyla, respectively, including various taxa without previously known MeHg-metabolizing microorganisms. The observed high diversity of putative MeHg-metabolizing microorganisms could largely be attributable to multiple independent horizontal gene transfer (HGT) events.

## RESULTS

**Concentrations of THg and MeHg in AMD sediments.** The concentrations of THg and MeHg in the 20 mine sites varied widely. The differences between the lowest concentrations (THg 4.28 $\mu$g kg$^{-1}$; MeHg 0.24 ng kg$^{-1}$) and the highest concentrations (THg 23,244 $\mu$g kg$^{-1}$; MeHg 1,188 ng kg$^{-1}$) reached up to nearly 5 orders of magnitude (Fig. 1B and C; Table S1). MeHg accounted for 0.06% to 3.78% of THg. Among the 20 mine sites, 4 (i.e., BPO, FAK, SKS, and LSA) had an average THg concentration of greater than 1,000 $\mu$g kg$^{-1}$, which were 2 to 10 times higher than those of the AMD sediments reported by He et al. (11). Another 4 sites (i.e., DAS, TTI, WUY, and ZHJ) had an average THg concentration ranging from 100 to 1,000 $\mu$g kg$^{-1}$, and the other 12 sites had an average THg concentration of less than 100 $\mu$g kg$^{-1}$. Consistent with the THg concentration results, high MeHg concentrations (>400 ng kg$^{-1}$ on average) were also observed in BPO, FAK, SKS, and LSA.

**Diversity, distribution, and abundance of putative MeHg producers.** A total of 46 dereplicated MAGs containing *hgcAB* were obtained in this study (Table S2). They belonged to 7 bacterial phyla (Fig. 2A): *Proteobacteria* (20: 19 *Deltaproteobacteria* and 1 *Betaproteobacteria*), *Nitrospirae* (15), *Firmicutes* (5), *Elusimicrobia* (2), *Spirochaetes* (2), *Actinobacteria* (1), and *Chloroflexi* (1). Among the 46 MAGs, 1 (i.e., BPO.bin3: 99.39% completeness and 0.04% contamination) (Table S2) was affiliated with the class *Betaproteobacteria*, which has not previously been implicated in Hg methylation (19). However, no recovered *hgcAB* genes from our metagenomic data set were affiliated with archaeal phyla (available at https://doi.org/10.6084/m9.figshare.21516258) (Fig. 2A; Fig. S1).

Most of the putative MeHg producers (42 out of 46) were ubiquitous in AMD sediments, as they could be observed individually in more than 50% of the samples (Fig. 2A). Notably, the MAG BPO.bin3, which was affiliated with a taxon without previously known MeHg producers, was observed in all of the samples. The relative abundances of three MAGs (i.e., BPO.bin3, FAK.bin3, and FAK.bin5) were >0.30%, respectively,

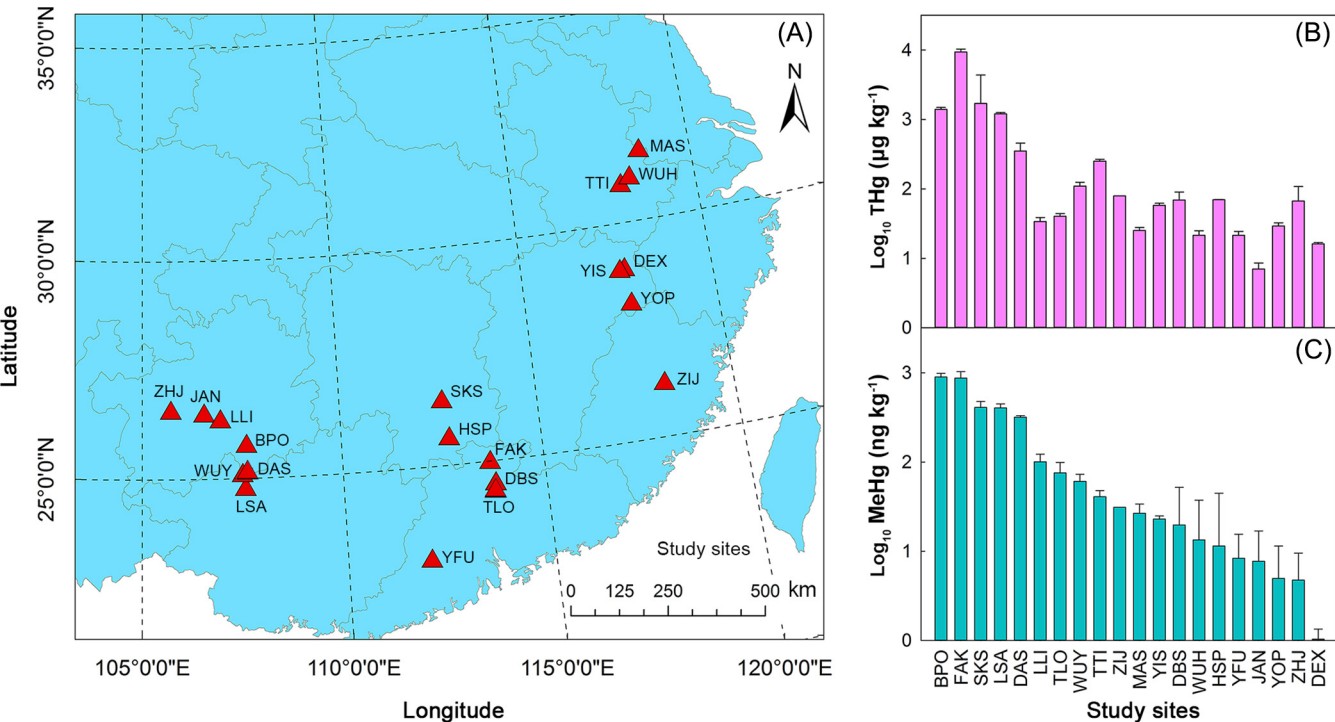

**FIG 1** Geographic location, total mercury (THg), and methymercury (MeHg) of the study sites. (A) Geographic locations of the 20 studied mine sites where we collected a total of 86 acid mine drainage (AMD) sediments. Abbreviations of the study sites are shown, and detailed information on these sites is provided in Table S1. (B) Concentrations (mean ± standard deviation) of THg in the studied AMD sediments. (C) Concentrations of MeHg in the studied AMD sediments. The $\log_{10}$-transformed concentration data are shown.

being higher than the average relative abundances of all of the *hgcAB*-containing MAGs across the samples (0.06%) (Fig. 2A). The total relative abundance of the putative MeHg producers in different sites varied from 0.15% to 7.32%, and the dominant putative MeHg producers in most of the mine sites were from *Deltaproteobacteria*, *Betaproteobacteria*, and *Nitrospirae* (Fig. 2B).

All of the *dsrAB* genes carried by the putative MeHg producers were of the reductive type (Fig. S2). They were detected in a considerable proportion of the *hgcAB*-containing MAGs belonging to *Deltaproteobacteria*, *Nitrospirae*, *Firmicutes*, and *Actinobacteria* (Fig. 2C; Table S3). Nearly 40% of the putative MeHg producers (17 out of 46) were identified as potential SRB (Fig. S3A; Table S3). They were affiliated with *Nitrospirae* (10), *Deltaproteobacteria* (5), and *Firmicutes* (2). Their total relative abundance in individual mine sites varied from 0.0019% to 3.07% (Fig. S3B). The two genes *omcF* and *omcS*, which are responsible for iron reduction, were found in five MAGs affiliated with *Nitrospirae* and in one MAG affiliated with *Firmicutes* (Fig. 2C; Table S3). The total relative abundances of these potential IRB in individual mine sites were generally much lower than those of the potential SRB (Fig. S3B). Only 4 putative MeHg producers had potentials of both sulfate reduction and iron reduction, and their total relative abundance values varied from 0.000013% to 1.30% (Fig. S3).

**Diversity, distribution, and abundance of MeHg degraders.** A total of 93 dereplicated MAGs carrying *merB* were recovered (Table S4), and these were distributed in eight phyla: *Actinobacteria* (32), *Proteobacteria* (23: 7 *Betaproteobacteria*, 7 *Gammaproteobacteria*, 5 *Acidithiobacillia*, 3 *Alphaproteobacteria*, 1 *Deltaproteobacteria*), *Firmicutes* (9), *Nitrospirae* (8), *Chloroflexi* (4), *Candidatus* Dormibacteraeota (2), *Euryarchaeota* (2), and *Planctomycetes* (1) (Fig. 3A). Remarkably, the putative MeHg degraders affiliated with *Planctomycetes*, *Candidatus* Dormibacteraeota, and the *Acidithiobacillia* class of *Proteobacteria* were first reported in this study. We could not identify genes encoding methanol dehydrogenase (i.e., the key enzyme putatively responsible for MeHg degradation in methanotrophs) (29).

The frequencies of occurrence of the *merB*-containing MAGs in all of the samples

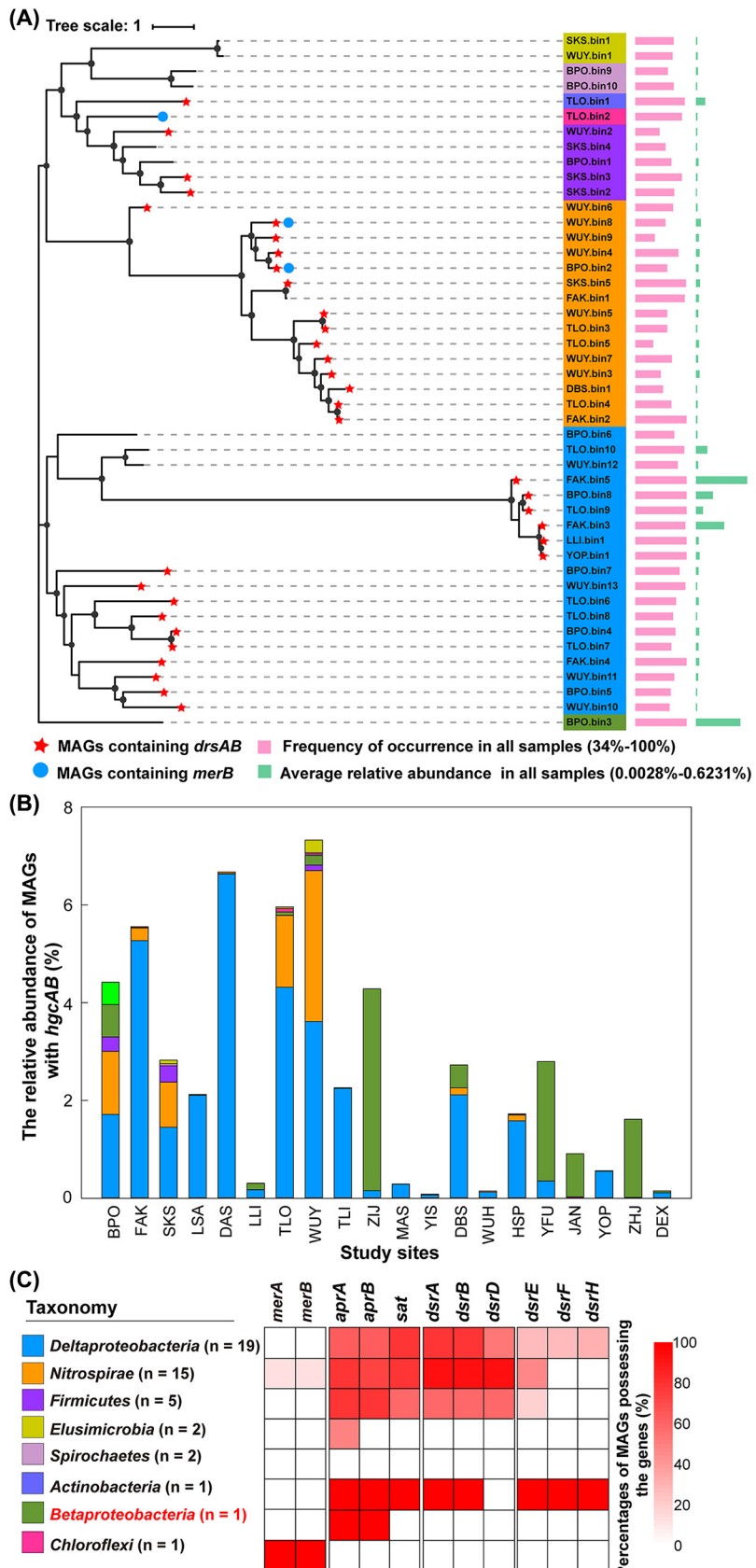

FIG 2 Analysis of the 46 good-quality or high-quality metagenome-assembled genomes (MAGs) of putative MeHg producers. (A) Phylogenic analysis of the 46 good-quality (completeness > 75% and

ranged from 58% to 100% (Fig. 3A). Most putative MeHg degraders exhibited an average relative abundance across sediments of less than 0.1%, except for 7 MAGs whose average relative abundances were >0.2%, individually. At the site level, the total relative abundances of putative MeHg degraders varied from 1.5% to 14% (Fig. 3B). The putative MeHg degraders affiliated with *Actinobacteria*, *Acidithiobacillia*, *Gammaproteobacteria*, and *Nitrospirae* were always dominant in most mine sites.

91 out of the 93 *merB*-containing MAGs carried *merA* genes (Fig. 3C; Table S5). In addition, 27 of these MAGs harbored other genes of the *mer* operon (Table S5), such as *merP* (encoding a periplasmic protein), *merT* (encoding inner membrane proteins involved in Hg (II) transport) and *merR* (encoding a regulatory protein). Furthermore, two MAGs (i.e., WUY.bin8 and BPO.bin2) belonging to *Nitrospirae* and one MAG (i.e., TLO.bin2) affiliated with *Chloroflexi* were found to contain both *hgcAB* and *merB* (Fig. 3C; Table S5), indicating that they were not only putative MeHg producers but also putative MeHg degraders.

*dsrAB* genes were detected in 18 putative MeHg degraders, including those of both the reductive type and the oxidative type (Table S5; Fig. S2). The putative MeHg degraders with reductive-type *dsrAB* were affiliated with *Nitrospirae* and *Firmicutes*, whereas those with oxidative-type *dsrAB* belonged to *Betaproteobacteria*, *Gammaproteobacteria*, and *Alphaproteobacteria* (Fig. 3C; Fig. S2). Although *Actinobacteria* harbored the greatest number of putative MeHg degraders, no *dsrAB* genes were identified in this phylum. *omcF* was detected in only two *merB*-containing MAGs (Table S5), suggesting that the iron reduction ability rarely occurred in the 93 putative MeHg degraders.

**HGT of *hgcAB* and *merB*.** We used a comparison of the concatenated HgcAB protein tree and its related genome-based phylogenetic tree (Fig. 4) to evaluate the extent to which HGT has influenced the organismal distribution of the *hgcAB* gene in the AMD sediments. A mismatching branching pattern was observed in the genome-based phylogenetic tree and in the HgcAB protein tree. Remarkably, deltaproteobacterial HgcAB sequences clustered with those affiliated with *Nitrospirae*, *Betaproteobacteria*, *Actinobacteria*, and *Spirochaetes*, indicating 10 independent HGT events. Additionally, *Chloroflexi*-affiliated HgcAB sequences were distant with those of other bacteria but were clustered with the reference euryarchaeotal HgcAB sequences, instead.

The MerB protein phylogeny also did not match well with its related genome-based tree phylogeny (Fig. 5). As the phylum harboring the greatest number of putative MeHg degraders, *Actinobacteria* likely acquired *merB* genes via 16 independent HGT events. Its MerB sequences clustered with those of *Firmicutes*, *Candidatus* Dormibacteraeota, *Betaproteobacteria*, and *Alphaproteobacteria*. Although the MerB sequences of other taxa were not as disparate as the *Actinobacteria* MerB sequences, they also clustered with the MerB sequences of other phyla. For example, the MerB sequences of *Nitrospirae* clustered with those of *Deltaproteobacteria*, *Actinobacteria*, and *Betaproteobacteria*, and those of *Chloroflexi* clustered with those of *Firmicutes*, *Actinobacteria*, *Euryarchaeota*, and *Gammaproteobacteria*. The only MerB sequences that clustered monophyletically among our data set were from *Acidithiobacillia* and *Candidatus* Dormibacteraeota.

**Key microbial and environmental factors influencing MeHg accumulation.** The relative abundances of the putative MeHg producers affiliated with *Deltaproteobacteria*,

**FIG 2** Legend (Continued)

contamination < 10%) or high-quality (completeness > 90% and contamination < 5%) *hgcAB*-carrying MAGs obtained in this study. The bootstrap values were based on 100 replicates, and those greater than 50% are marked with black circles. The taxonomy of individual MAGs is generally shown at the phylum level, with those affiliated with *Proteobacteria* being shown at the class level. Each of the eight taxonomic lineages is indicated by a separate color, as shown in panel C. A taxonomic lineage without previously known MeHg producers is marked in red font. The number of MAGs belonging to each of the eight taxonomic lineages is also given at the bracket following the name of that lineage. MAGs containing *dsrAB* and *merB* are indicated with red stars and blue circles, respectively. The frequencies of occurrence of the individual MAGs in all samples as well as their average relative abundances are indicated on the right of the phylogenetic tree with pink and green bars, respectively. (B) The relative abundance of the MAGs affiliated with each of the eight taxonomic lineages in each study site. The color coding for the taxonomic lineages is the same as that in panel C. (C) Distribution of *merA*, *merB*, and sulfate reduction-associated genes in the MAGs affiliated with each of the eight taxonomic lineages.

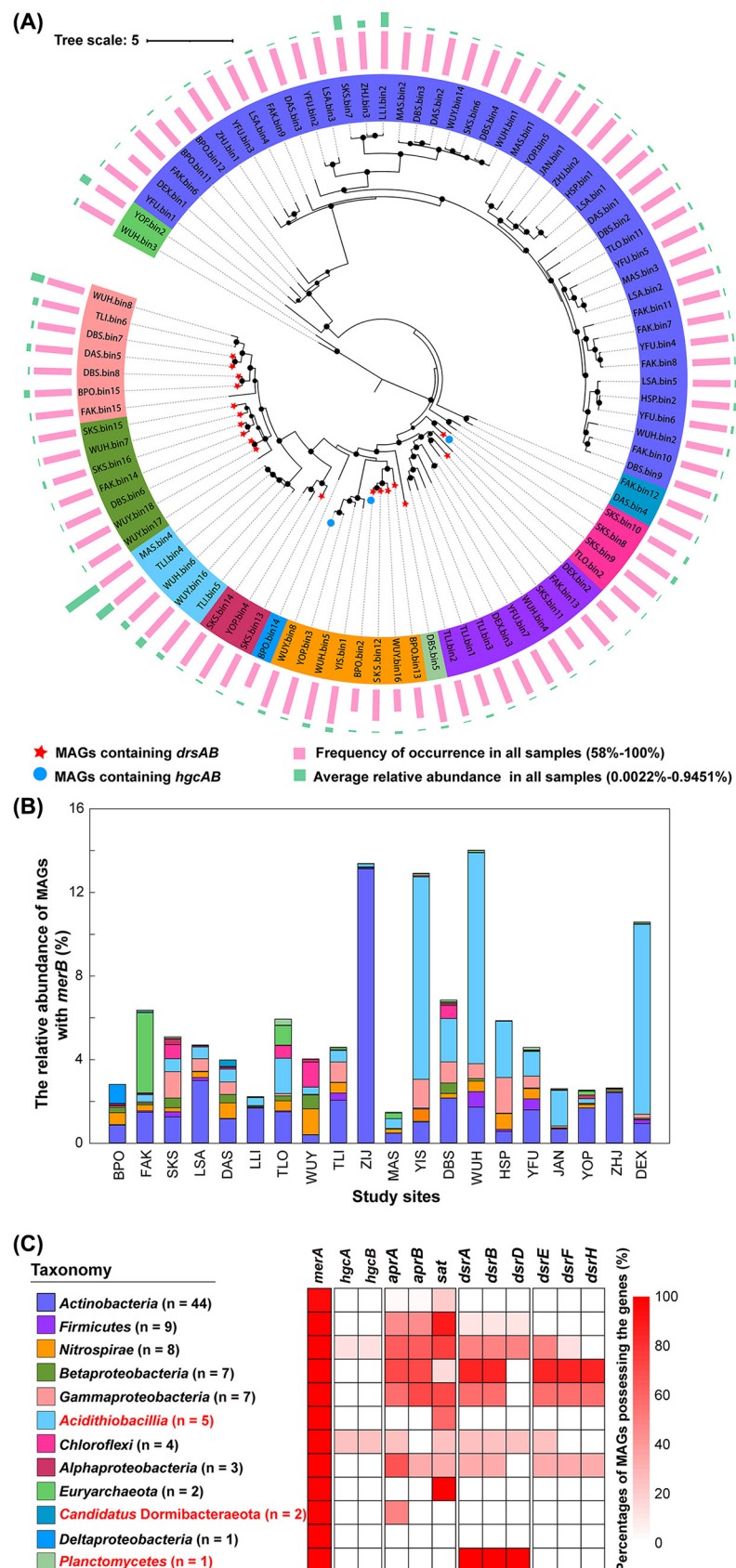

FIG 3 Analysis of the 93 good-quality or high-quality MAGs of putative MeHg degraders. (A) Phylogenic analysis of the *merB*-carrying good-quality MAGs obtained in this study. Bootstrap values were based on

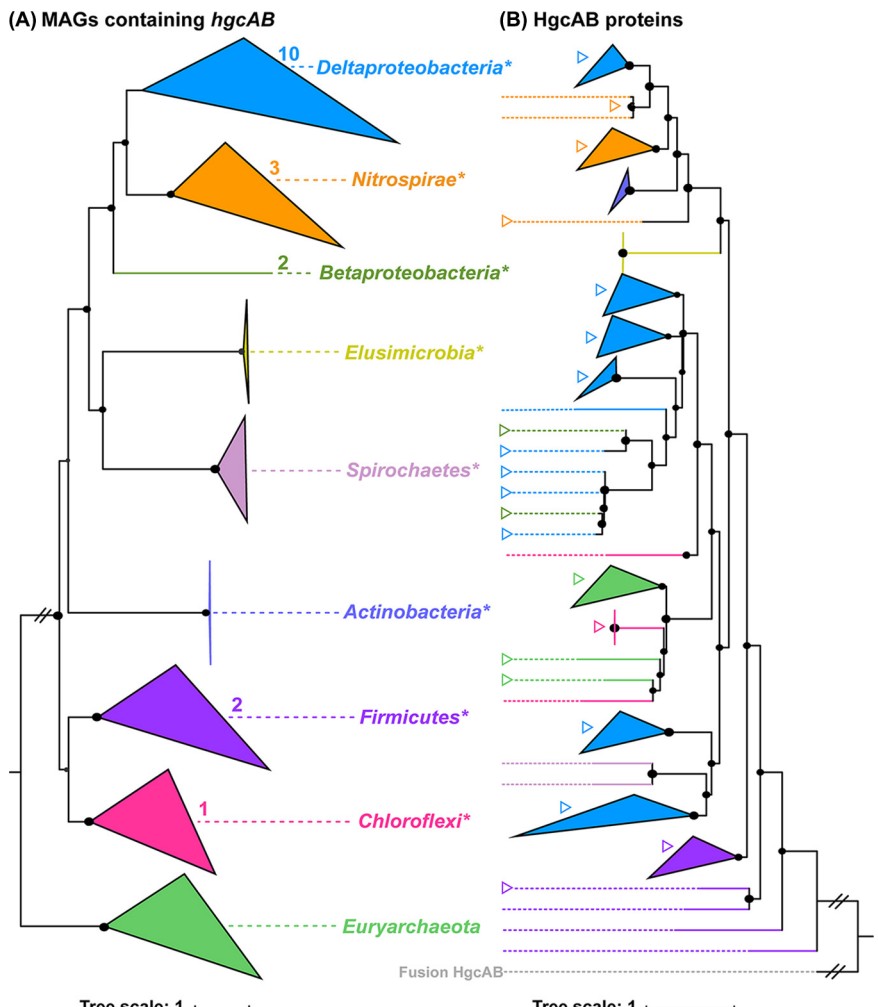

**FIG 4** Comparison of the genome-based phylogenetic tree and concatenated HgcAB protein tree for putative MeHg producers. (A) Phylogenic analysis of *hgcAB*-carrying MAGs and the reference genome. (B) Phylogenic analysis of the concatenated HgcAB proteins and the reference HgcAB proteins. Sequences are generally grouped at the phylum level, with those of *Proteobacteria* being grouped at the class level. Each phylum or class is indicated by a separate color to identify horizontal gene transfer (HGT), based on inconsistent branching patterns. The numbers in panel A represent the total numbers of independent HGT events associated with putative *hgcAB* genes in specific phyla or classes in AMD sediments. The right triangles in panel B represent the HGT events of putative HgcAB in specific phyla or classes in AMD sediments. The bootstrap values were based on 100 replicates, and those greater than 50% are shown with black circles. Asterisks indicate the branches obtained in this study that contained *hgcAB* genes. The HgcAB protein tree was rerooted with fusion HgcAB of *Streptomyces* sp. CNQ-509.

*Nitrospirae*, and *Firmicute*s were significantly ($P < 0.05$) positively correlated with the concentration of MeHg in the AMD sediments (Fig. 6A–C). In contrast, the relative abundance of the putative MeHg degraders affiliated with *Acidithiobacillia* was significantly negatively correlated with the MeHg concentration (Fig. 6D). Among the 14 selected

**FIG 3** Legend (Continued)

100 replicates, and those greater than 50% are marked with black circles. The taxonomy of individual MAGs is generally shown at the phylum level, with those affiliated with *Proteobacteria* being shown at the class level. Each of the 12 taxonomic lineages is indicated by a separate color, as shown in panel C. Three phyla without previously known MeHg degraders are marked in red font. The number of MAGs belonging to each of the 12 taxonomic lineages is also given at the bracket following the name of that lineage. MAGs containing *dsrAB* and *hgcAB* are indicated with red stars and blue circles, respectively. The frequencies of occurrence of individual MAGs in all samples as well as their relative abundances are indicated on the right of the phylogenetic tree with pink and green bars, respectively. (B) Relative abundance of the MAGs affiliated with each of the 12 taxonomic lineages in each study site. The color coding for the taxonomic lineages is the same as that in panel C. (C) Distribution of *merA*, *hgcAB*, and sulfate reduction-associated genes in the MAGs affiliated with each of the 12 taxonomic lineages.

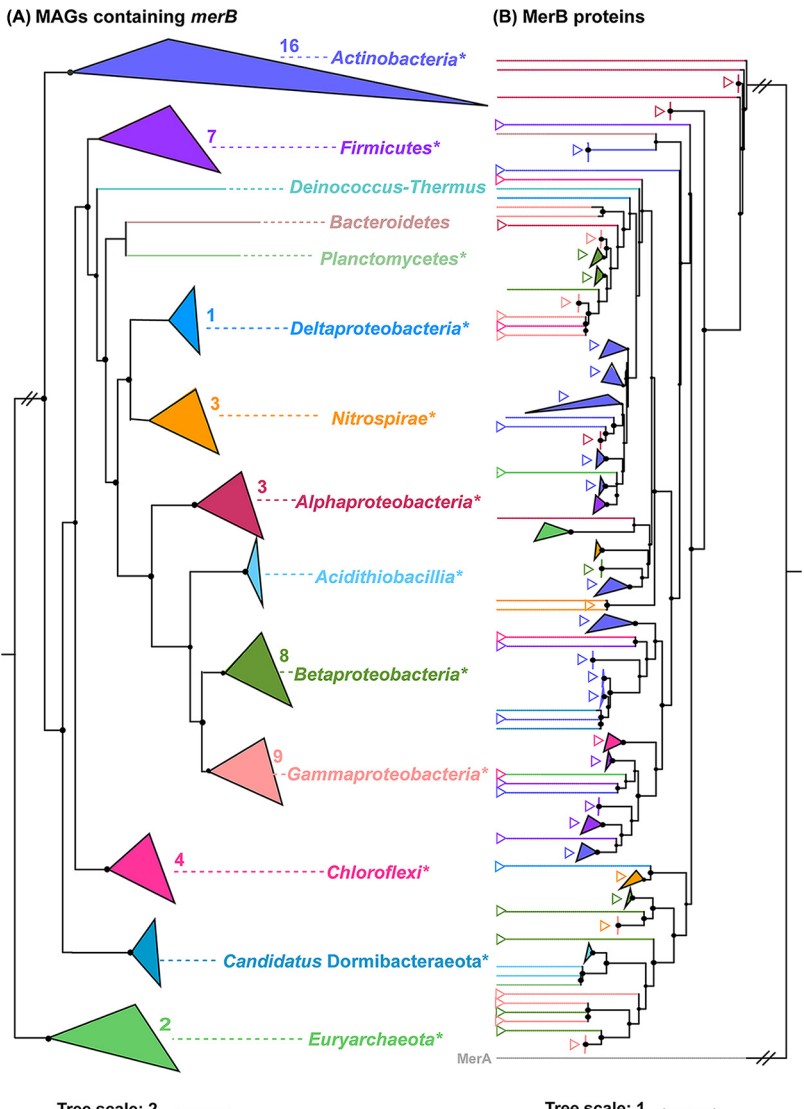

**FIG 5** Comparison of the genome-based phylogenetic tree and the MerB protein tree for putative MeHg degraders. (A) Phylogenic analysis of *merB*-containing MAGs and the reference genome. (B) Phylogenic analysis of putative MerB proteins and the reference MerB proteins. Sequences are generally grouped at the phylum level, with those of *Proteobacteria* being grouped at the class level. Each phylum or class is indicated by a separate color to identify HGT, based on inconsistent branching patterns. The numbers in panel A represent the total numbers of independent HGT events associated with putative *merB* genes in specific phyla or classes in AMD sediments. The right triangles in panel B represent the HGT events of putative MerB in specific phyla or classes in AMD sediments. The bootstrap values were based on 100 replicates, and those greater than 50% are shown with black circles. Asterisks indicate the branches obtained in this study that contained *merB* genes. The MerB protein tree was rerooted with MerA of *Bacillus* sp. RC607.

environmental factors (Table S1), only the total carbon (TC) content and the $Fe^{2+}/Fe^{3+}$ ratio were significantly ($P < 0.05$) positively correlated with the MeHg concentration (Fig. 6E and F).

Multiple linear regression (MLR) was further used to determine the contributions of the six factors identified by the correlation analysis in explaining MeHg accumulation. The relative abundance of *Nitrospirae*-related putative MeHg producers was removed due to its low contribution during the best subsets screening, which was based on the Akaike information criterion (AIC). The best model of MLR with five factors explained 62.5% of the variance of the accumulation of MeHg in the AMD sediments (Fig. 6G; Table S6). Specifically, the relative abundance of deltaproteobacterial putative MeHg

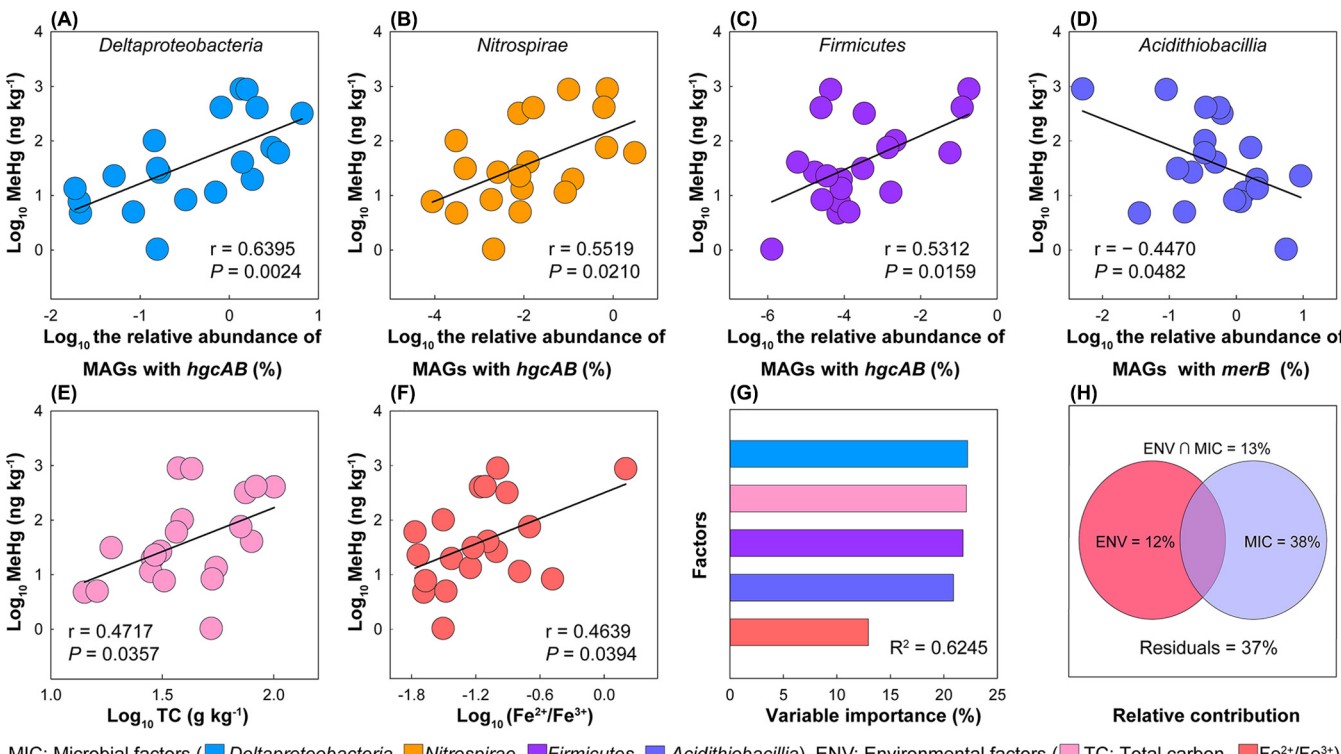

FIG 6 Key microbial and environmental factors correlating with MeHg accumulation in AMD sediments. (A–C) Pearson correlations between the relative abundances of putative MeHg producers affiliated with *Deltaproteobacteria*, *Nitrospirae*, or *Firmicutes* and the MeHg concentrations. (D) Pearson correlations between the relative abundance of putative MeHg degraders affiliated with *Acidithiobacillia* and the MeHg concentrations. (E and F) Pearson correlations between the total carbon (TC) content or the $Fe^{2+}/Fe^{3+}$ ratio and the MeHg concentrations. (G) The relative importance of individual factors in explaining MeHg accumulation, as assessed by a multiple linear regression model. (H) The relative importance of the key microbial and environmental factors contributing to MeHg accumulation, as assessed by a variance decomposition analysis. The mean values of the MeHg concentrations and the microbial and environmental factors of the individual study sites were used in the analysis.

producers and the TC content contributed to 22.2% and 22.1% of the variable importance, respectively, followed by the relative abundances of the *Firmicutes*-related putative MeHg producers and the *Acidithiobacillia*-related putative MeHg degraders as well as the $Fe^{2+}/Fe^{3+}$ ratio. In addition, our variance decomposition analysis also revealed that the five factors together explained 63% of the variance of the accumulation of MeHg, with the microbial factors alone explaining as much as 38% of the variance of the accumulation of MeHg (Fig. 6H).

**Metabolic profiles of the putative MeHg producers and degraders.** Gene-level metabolic profiles were analyzed for the 46 putative MeHg producers and for the 93 putative degraders. Our results showed that there were few differences among these MeHg-metabolizing microorganisms in terms of their metabolic potential for autotrophic carbon fixation (Fig. S4), phosphorous cycling (Fig. S5), sulfur oxidation (Fig. S6), low pH adaption, and low oxygen tolerance (Fig. S7).

When the putative MeHg producers were taken into account, the genes involved in the metabolism of nitrogen (N) were mainly detected in the MAGs belonging to *Deltaproteobacteria*, *Nitrospirae*, and *Firmicutes* (Fig. 7). Particularly, the N fixation-associated genes *nifHDK* and their cofactor genes *nifNEX* were found in a large proportion of the *hgcAB*-containing MAGs that were affiliated with *Deltaproteobacteria* (68.42%), *Nitrospirae* (80%), and *Firmicutes* (80%). In contrast, these genes were absent in most of the MAGs of the other MeHg producer taxa. A similar pattern was also observed for the genes responsible for copper (Cu) tolerance and cobalt-zinc-cadmium (Co-Zn-Cd) resistance.

Compared to the putative MeHg producers, a higher proportion of the putative MeHg degrader taxa carried the genes involved in N metabolism and in Cu and Co-Zn-Cd resistance (Fig. 7), although these two contrasting MeHg-metabolizing microbial groups

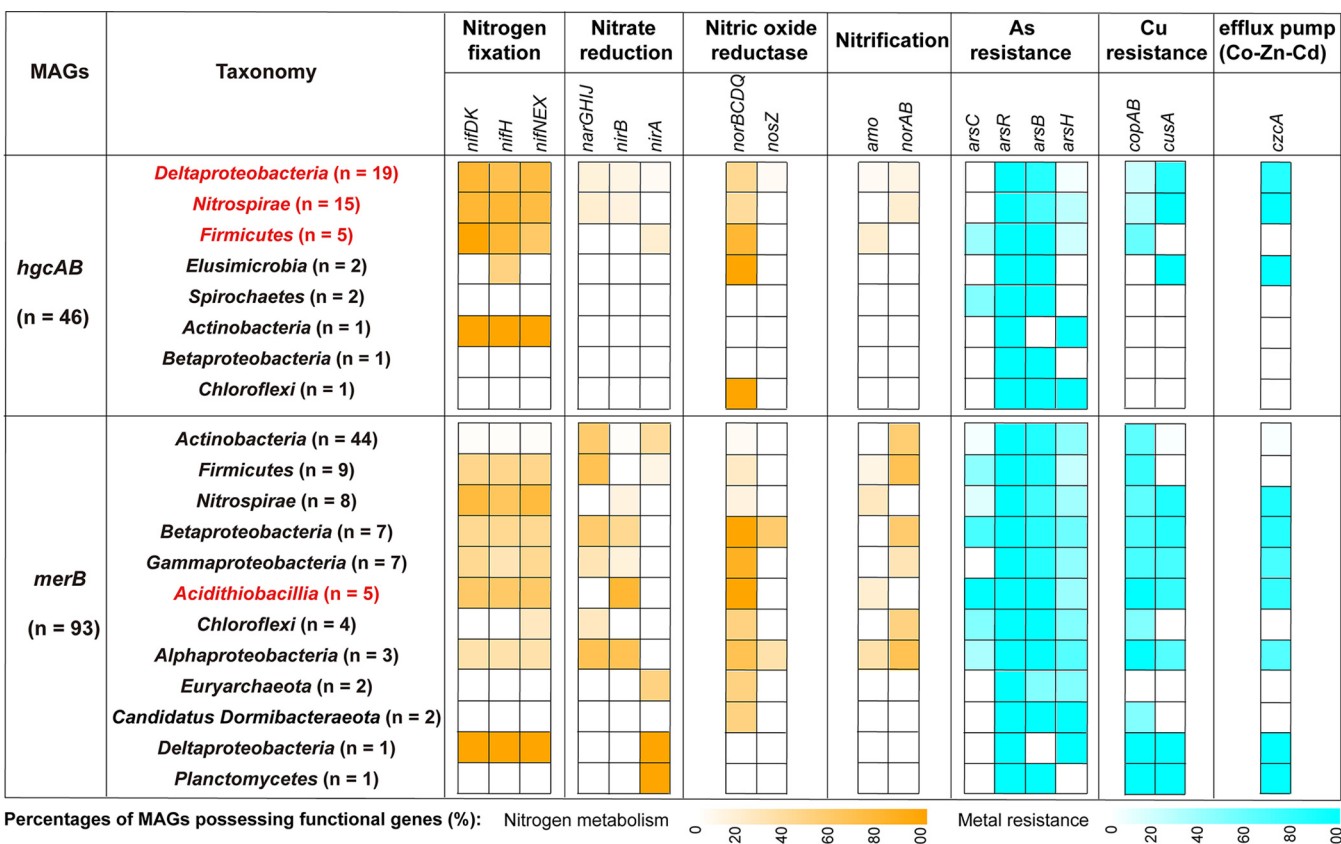

**FIG 7** Distribution of the genes responsible for nitrogen metabolism and metal tolerance in putative MeHg producers and degraders. The taxonomy of individual MAGs is generally shown at the phylum level, with those affiliated with *Proteobacteria* being shown at the class level. Four taxonomic lineages that were identified as important determinants of MeHg accumulation in AMD sediments are marked in red font. The color of each cell refers to the percentage of MAGs affiliated with each taxonomic lineage containing the gene(s) involved in the individual pathways of nitrogen metabolism or metal tolerance. As, arsenic; Cd, cadmium; Co, cobalt; Cu, copper; Zn, zinc.

exhibited similar metabolic potential for arsenic (As) tolerance. Notably, 100%, 100%, and 80% of the *merB*-containing MAGs that are affiliated with *Acidithiobacillia* harbored the genes responsible for As, Cu, and Co-Zn-Cd resistance, respectively, which were always higher than those of the other 11 putative MeHg degrader taxa (Fig. 7).

## DISCUSSION

**Diverse putative MeHg producers and degraders in AMD sediments.** MeHg accumulation in the environment is determined largely by the balance between MeHg production and degradation (33, 34), which are driven mainly by microorganisms (21). Therefore, exploring the diversities of not only microbial MeHg producers but also microbial MeHg degraders is an important step toward a deep understanding of the constraints on MeHg accumulation in the environment. However, few previous studies have done so. In order to narrow this critical knowledge gap, we selected AMD sediments as a model ecosystem. There are three major reasons for our selection. First, AMD sediments are often considered a MeHg hot spot, which was also exemplified by our observation that the highest concentration of MeHg in the AMD sediments from southern China was up to 1,188 ng kg$^{-1}$ (Fig. 1C; Table S1), being nearly four times more than that of some freshwater sediments (9, 10). Second, the diversities of microbial MeHg producers and degraders in AMD sediments have not yet been studied. Third, the reduced complexity of the microbial communities in AMD sediments is likely to facilitate the recovery of quality MAGs from metagenomes (35).

Taking advantage of publicly available isolate genomes, MAGs, and newly generated metagenomes, McDaniel et al. (19) recently demonstrated that putative MeHg producers

from various ecosystems span up to 30 phyla. In this study, all MAGs of putative MeHg producers were affiliated with bacterial phyla (Fig. 2A), suggesting that bacterial MeHg producers could play a more important role than do archaea in MeHg production in AMD sediments. Compared with the taxonomic affiliation that is based on the National Center for Biotechonlogy Information (NCBI) taxonomy, all of the Hg-methylating MAGs in this study belonged to phyla with previously known MeHg producers. However, at the class level, we fou;nd *Betaproteobacteria*, for the first time, as a class harboring putative MeHg producers (Fig. 2A; Fig. S8). Moreover, compared with the Genome Taxonomy Database (GTDB)-based results of most recent studies (19, 20, 36), more than one-half of the 46 *hgcAB*-containing MAGs were affiliated with several distinct lineages that were not previously reported to harbor *hgcAB*. They included one uncultivated candidate phylum (SZUA-79), four classes (UBA6077 in *Chloroflexota*, *Dissulfuribacteria* in *Desulfobacterota*, *Thermincolia* in *Firmicutes*, and *Gammaproteobacteria* in *Proteobacteria*), two orders (BMS3ABIN01 in *Actinobacteriota* and *Ammonifexales* in *Firmicutes*), two families (JdFR-88 in *Nitrospirota* and UBA8932 in *Spirochaetota*), and one genus (*Sulfobium* in *Nitrospirota*) (Table S2).

Remarkably, one putative MeHg producer of this study (i.e., BPO.bin3 affiliated with *Betaproteobacteria*) contained two pairs of *hgcAB* genes (Fig. 3; Fig. S1). Similarly, we also observed the presence of two pairs of *hgcAB* genes in 10 previously reported MAGs (completeness > 75% and contamination < 10%) (Table S7), which were recovered from various aquatic environments (18–20). More importantly, complete genomes of four isolates affiliated with *Deltaproteobacteria* and *Firmicutes* were found to possess two pairs of *hgcAB* genes (Table S7). Although such a phenomenon has received little attention in the literature, it raises an interesting question of whether the microbes with two pairs of *hgcAB* have a greater Hg-methylating capacity than do those with only one pair of *hgcAB*.

Compared to MeHg producers, the diversity of MeHg degraders remains poorly understood, as evidenced by the fact that the *merB* gene is currently identified in only five strains and one MAG affiliated with two phyla (i.e., *Firmicutes* and *Proteobacteria*) (24–26). In this study, the putative MeHg degraders belonged to eight phyla, among which six had not previously been reported to display MeHg-demethylation ability. In comparison with the MerB sequences in the latest protein families (Pfam) database, NCBI GenBank, and previous studies (24, 25, 37), eight MAGs obtained in this study that contained the *merB* gene were from three taxa without previously known MeHg degraders (Fig. 3; Fig. S9), including two phyla (*Planctomycetes* and *Candidatus* Dormibacteraeota) and one class (*Acidithiobacillia*). Nonetheless, our results have greatly improved our understanding of the phylogenetic diversity of putative MeHg degraders. Note also that the average occurrence frequencies of the putative MeHg producers and degraders from these taxa were greater than in 86% of the samples (Table S8), indicating that they were widespread in the AMD sediments.

It is noteworthy that two MAGs belonging to *Nitrospirae* and one MAG belonging to *Chloroflexi* contained both *hgcAB* and *merB* (Fig. 3). This meant that they could be both putative MeHg producers and degraders. A similar phenomenon was also observed in an unbinned scaffold from a hot spring (26). Furthermore, *Geobacter bemidjiensis* Bem isolated from a petroleum-contaminated aquifer has been the only one strain that was verified to have the ability to simultaneously methylate Hg and degrade MeHg (38).

**The coupling of the Hg cycle and the iron-sulfur cycle in AMD sediments.** Sulfate reduction can provide the driving force for the biological methylation of Hg (39). Sulfide generated via sulfate reduction is a strong inorganic ligand that can bind with Hg, and some sulfur-Hg complexes have been identified as important substances for Hg methylation (40). In addition, the IRB-mediated iron reduction process is often coupled with the biogeochemical cycle of sulfur. This kind of coupling often produces iron-sulfur compounds that can adsorb Hg and can thereby affect the production of MeHg (41, 42). Therefore, in sulfate-rich environments, SRB usually account for a large proportion of the MeHg producers, followed by IRB or others (32, 43). In this study, 17

of the 46 putative MeHg producers were identified as SRB, with this number being much higher than the percentage reported by Peterson et al. (44), who observed that only 5 of the 108 putative MeHg producers from a sulfate-enriched lake were SRB. Such a discrepancy could be partly attributed to the much lower level of sulfate in the lake compared to those in the AMD sediments (on average: 0.02 versus 14.3 g kg$^{-1}$). Despite this, the relative abundance of the putative MeHg producers from SRB, on average, accounted for 17.1% of the 46 putative MeHg producers (Fig. S3B), being slightly lower than that (i.e., 22%) recorded by Peterson et al. (44). In addition, the number of putative MeHg producers from the IRB and their relative abundance in the AMD sediments were much lower than those from the SRB (Fig. S3). Nonetheless, neither the relative abundance of the *hgcAB*-carrying SRB nor that of the *hgcAB*-carrying IRB was significantly correlated with the concentration of MeHg in the AMD sediments (Fig. S3), suggesting that these two subgroups of putative MeHg producers were not the only contributors to MeHg accumulation.

Among the currently known MeHg degraders, those with a non-*merB*-mediated pathway often have sulfate reduction abilities (27, 45). However, the sulfate and iron reduction abilities of the MeHg degraders with a *merB*-mediated pathway are still poorly understood. So far, only one strain of the MeHg degrader carrying *merB* (*G. bemidjiensis* Bem) has been reported to have the ability to reduce iron (38). In this study, one and two putative MeHg degraders with *merB* were identified to be SRB and IRB, respectively (Fig. 3C; Tables S3 and S5). However, as their relative abundances were evidently lower than those of most of the other putative MeHg degraders, these three putative MeHg degraders likely played a minor role in the accumulation of MeHg in the AMD sediments (Fig. S3).

**HGT of *hgcAB* and *merB* in microorganisms in AMD sediments.** As a major evolutionary force for prokaryotes, HGT plays an important role for microorganisms that are adapting to new ecological niches (46). A recent study based on nearly 1,000 (putative) MeHg producers from diverse ecosystems has revealed that the HGT of *hgcAB* genes occurred mainly in *Deltaproteobacteria* and *Firmicutes* (19). For a better comparison, the putative HGT events reported by McDaniel et al. (19) were recalculated according to the method of Anantharaman et al. (47) (i.e., the same method that was employed for the MAGs of this study). In partial agreement with the findings of McDaniel et al. (19), our results showed that a large proportion of the HGT events of *hgcAB* genes in AMD sediments were observed in *Deltaproteobacteria* (Fig. 4). Notably, up to 10 independent HGT events of *hgcAB* genes occurred in *Deltaproteobacteria* in AMD sediments, and this number was greater than the number that was recently reported for the same taxon in other habitats (19). The *hgcAB*-containing MAGs of *Deltaproteobacteria* occurred in more than 98% of the AMD sediments (Table S8). These findings indicate that the evolutionary history of *hgcAB* genes in *Deltaproteobacteria* could be complex. This kind of complexity could be at least partially attributable to the harsh environmental conditions in AMD ecosystems (48).

The HGT events of *merA* were reported to have occurred in several phyla, including *Euryarchaeota*, *Aquificae*, *Crenarchaeota*, *Proteobacteria*, and *Firmicutes* (37). However, to our knowledge, the influence of HGT events on the phylogenetic distribution of the *merB* gene has not been previously explored. Our results showed that HGT events of *merB* occurred in as many as seven bacterial taxa, including *Actinobacteria*, *Alphaproteobacteria*, *Betaproteobacteria*, *Gammaproteobacteria*, *Firmicutes*, and *Chloroflexi* (Fig. 5). Among them, *Actinobacteria* contributed more than did other taxa to the total number of the HGT events, with 65.8% of the total putative MeHg degraders from this taxon being involved. In contrast, no HGT events of *merB* or *hgcAB* were observed in the two *Nitrospirae*-like MAGs that were carrying both *hgcAB* and *merB*. As to another putative MeHg producer and degrader from *Chloroflexi*, its *merB* clustered with that of *Firmicutes*, and its *hgcAB* clustered with the reference sequences from *Euryarchaeota*, indicating that both the *merB* and the *hgcAB* of this MAG were obtained through lateral gene transfer events. These findings raise the possibility that MeHg production and degradation are relatively ancient traits that could occur in

the same species. However, at this stage, we are not able to identify the specific taxon in which MeHg producers and/or degraders first appeared.

**Factors affecting MeHg accumulation in AMD sediments.** Recently, Lin et al. (20) demonstrated that a previously unrecognized MeHg-producing phylum (i.e., *Marinimicrobia*) played a more important role than did the other six phyla that carry *hgcAB* in Hg methylation in the waters of Saanich Inlet. This interesting finding inspired us to explore the relative importance of the observed putative MeHg producers and degraders belonging to individual phyla (but to classes in *Proteobacteria*) in explaining the variance of the accumulation of MeHg in AMD sediments. Among the seven putative MeHg producer phyla (including two classes in *Proteobacteria*) and eight putative MeHg degrader phyla (including five classes in *Proteobacteria*) that inhabited the AMD sediments, only four taxa were found to be closely correlated with MeHg accumulation (Fig. 6A–D). More specifically, the putative MeHg producers from *Deltaproteobacteria*, *Nitrospirae*, and *Firmicutes* were positively correlated with the concentration of MeHg, whereas those putative MeHg degraders from *Acidithiobacillia* were negatively correlated with the concentration of MeHg. All of the members of the four taxa possessed genes involved in low pH adaptation, reflecting their potential to survive and thrive in extremely acidic AMD sediments (Fig. S7). Moreover, our results showed that a higher proportion of the members of these four taxa have the metabolic potential for N cycling (particularly N fixation) and metal tolerance (particularly Cu tolerance), compared with those of the other taxa (Fig. 7). Bioavailable N is usually a limiting factor for microbes to grow in AMD ecosystems (49). Therefore, N fixation ability is likely an important source of competitive advantage for microbes. Likewise, the elevated Cu tolerance of certain microbes can help them to survive in the metal-enriched AMD sediments. The MLR analysis and variance decomposition analysis showed that *Nitrospirae* had a smaller ability in explaining the variance of the accumulation of MeHg in AMD sediments than did the other three taxa (Fig. 6G and H). A possible explanation is that *Nitrospirae* harbors both putative MeHg producers and degraders, although other potential reasons remain unclear.

As a first attempt to compare the relative importance of biotic and abiotic factors in explaining the variance of the accumulation of MeHg, this study showed that the TC content and $Fe^{2+}/Fe^{3+}$ ratio were two important abiotic determinants (Fig. 6E–H). Under oligotrophic conditions, a lower TC content often reflects a reduced availability of carbon (C) sources (50). A shortage of C sources likely results in a reduction in the growth of microbes (including those with Hg-methylation abilities). This can explain why the TC content has a positive correlation with the accumulation of MeHg (Fig. 6E). In AMD ecosystems with pH values below 3.5, $Fe^{3+}$ is a major oxidant (51). Thus, a high $Fe^{2+}/Fe^{3+}$ ratio indicates a low redox state in acidic environments (52). Our observation that MeHg accumulation in AMD sediments increased with an increasing $Fe^{2+}/Fe^{3+}$ ratio was consistent with those of previous studies, which showed that Hg methylation was favored at a low redox state in estuarine sediments (53, 54). Apart from enhancing Hg-methylation activity (55), a low redox state can also improve MeHg accumulation by inhibiting the demethylation process (56). Additionally, the THg was also observed to have a significant correlation with MeHg (Fig. S10). Inorganic Hg is the substrate of MeHg and often constitutes the majority of the THg in various environments (6). In the AMD sediments, however, MeHg accounted for only 0.0001% to 0.76% of the THg, indicating that the substrate concentration (i.e., inorganic Hg) was unlikely to be a limiting factor of the production of MeHg (57). Collectively, our results regarding the effects of abiotic factors suggest that decreasing C input and improving aeration are beneficial to the mitigation of MeHg accumulation in AMD sediments.

**Conclusions.** Using AMD sediments as a model ecosystem, this study revealed that the diversity of putative MeHg-metabolizing microorganisms (particularly MeHg degraders) was much higher than was previously recognized and that this great diversity has been formed largely via HGT events. Meanwhile, we demonstrated the remarkable overall ability of five ecological factors (including two taxa of putative MeHg producers, one taxon of putative MeHg degraders and two abiotic factors) that were closely correlated with the MeHg concentration in the AMD sediments to explain the

variance of the accumulation of MeHg in such habitats across a large spatial scale. These findings can improve our understanding of the ecology of putative MeHg-metabolizing microorganisms and can guide us in the development of effective management strategies for reducing MeHg accumulation in AMD sediments. Nevertheless, this study is based on metagenomics, and the relative abundances of putative MeHg producers/degraders may not necessarily equate with their actual capacities of methylation/demethylation. Therefore, more studies employing other experimental approaches (e.g., metatranscriptomics) are needed in order to verify the findings observed in this study.

## MATERIALS AND METHODS

**Site description and sample collection.** The 20 mine sites under study are located in seven provinces across southern China (Fig. 1A; Table S1). The latitude and longitude of these mine sites ranged from 22.96°N to 31.68°N and from 105.73°E to 118.63°E, respectively (Table S1). They cover an area of approximately 500,000 km². At each mine site, 3 to 10 AMD sediments (250 mL of mud each) were collected from an AMD pond using a sediment collector at a depth of $5 \pm 1$ cm below the water-sediment interface during July and August of 2017. The sample size of a given AMD pond was approximately proportional to its area. Each sample was divided into two parts. One part (approximately 50 mL) was placed in a 50 mL sterile tube, transported back to the laboratory in a thermic box with dry ice, and stored at −80°C for DNA extraction. The other part (approximately 200 mL) was transported back at room temperature, air-dried, and sieved with mesh sizes of 100 to 150 for physicochemical analyses.

**Physicochemical analyses.** The concentrations of THg and MeHg in the individual samples were determined at the Institute of Geochemistry, Chinese Academy of Sciences (Guiyang, China). To determine the THg concentrations, 0.2 g of the dried samples were digested by freshly mixed analytical grade $HCl/HNO_3$ (1:3, vol/vol) in a water bath at 95°C, and they were subsequently analyzed via cold vapor atomic fluorescence spectrometry (CVAFS; Tekran 2500, Tekran Inc., Canada) as described previously) (58). MeHg was extracted from 0.4 g of the dried samples using $CuSO_4$-methanol/solvent extraction, and it was then quantified via gas chromatography (GC) (CVAFS; Tekran 2700, Tekran Inc., Canada) as described previously (58). All of the samples were also analyzed by standard methods for 12 other physicochemical parameters, including pH, total carbon, and ferrous and ferric iron ($Fe^{2+}$ and $Fe^{3+}$). More details are provided in the supplemental material.

**DNA extraction and metagenomic sequencing.** The total microbial genomic DNA was extracted from 1 to 5 g of AMD sediment samples using a FastDNA Spin Kit (MP Biomedicals, USA) according to the manufacturer's instructions. For each sample, multiple extractions were performed, using 500 mg of sediment as the input for one extraction. Then, the DNA from the multiple extractions were combined. The extracted DNA yield and purity were evaluated using a NanoDrop 2000 spectrophotometer (Thermo Scientific, USA). The DNA library of each sample was constructed using a NEBNext Ultra II DNA Prep Kit (New England Biolabs, USA) following the manufacturer's instructions. A total of 86 DNA libraries were successfully constructed, and they were sequenced on an Illumina HiSeq 2500 platform in the PE150 mode (Illumina, USA).

**Metagenomic assembly and genome binning.** High-quality (HQ) reads were generated by in-house pipelines from our metagenomic data sets, including the removal of duplicated reads, low-quality reads (Q30), and reads with more than 5 ambiguous "N" bases (59). The HQ reads from each sample were individually assembled into scaffolds using SPAdes (version 3.9.0) with the parameters "-k 21, 33, 55, 77, 99, 127 –meta" (60). Only the scaffolds with lengths of ≥2,000 bp were retained for genome binning with the DAS Tool (version 1.00) (61), including four binning methods (ABAWACA v.1.00, CONCOCT v.04.0, MaxBin v2.2.2, and MetaBAT v.2.12.1) (62–65). The resulting MAGs were further improved using RefineM (version 0.0.25), and the scaffolds were manually removed with conflicting phylum-level taxonomy (66). The completeness, contamination, and strain heterogeneity values of the refined MAGs were assessed using CheckM v1.0.12 (67). Good-quality and high-quality MAGs (>75% completeness and <10% contamination) (68) were selected for further analysis.

**Identification of *hgcAB*, *merB*, and other functional genes in MAGs.** Genes were predicted by Prodigal (version 1.0.2) (69). The *hgcA* and *hgcB* genes are indispensable for the occurrences of Hg methylation that have been confirmed in known MeHg producers, but they do not need to be located contiguously within the genomes (14). Hidden Markov models (HMMs) were built using reference HgcA and HgcB sequences (Table S9) from previous studies (14, 70) using HMMER (version 3.2) (71), and they were used to screen the homologous sequences from the MAGs. Only the sequences with a maximum E value of 10e−40 were aligned with the highly conserved "G(I/V)NVWCA(A/G)" motif of HgcA (Fig. S1 and S11) (15, 69) using MEGA (version 7.0.26). The sequences with conserved domains of HgcA were matched with MAGs for the further screening of HgcB. The HgcB sequences were identified by matching the conserved domains (two strictly conserved $CX_2CX_2CX_3C$ motifs) (Fig. S12) with the E values that were <10e−5 (15). The 70 good-quality and high-quality MAGs that contained both the *hgcA* and *hgcB* genes were identified as putative MeHg producers in the AMD sediment samples.

The *merB* gene encoding organomercury lyase possesses two highly conserved cysteines at positions 96 and 159 (numbering based on MerB from *Escherichia coli*) (Fig. S13) (72). Based on these conserved positions, we chose 41 MerB sequences (Table S10) from a previous study (37) and the Pfam database (accession number 03243) to build an HMM for the screening of MerB from our MAGs. The *merA* gene,

which encodes a mercuric reductase, was screened from our MAGs using an HMM search that was built using known MerA sequences from a previous study (37). In addition, other genes of the *mer* operon were identified, and these are elaborated in the supplemental material.

All of the protein-coding genes of the MAGs containing *hgcAB* or *merB* were searched against the eggNOG and KEGG databases for functional annotations using DIAMOND (version 2.0.4.142) with an E value threshold of 10e−5. According to Anantharaman et al. (47), the SRB that are responsible for the reduction of sulfate to sulfide should have six core genes (i.e., *aprA*, *aprB*, *sat*, reductive *dsrA* and *dsrB*, and *dsrD*) but should not contain three genes of the *dsr* operon (i.e., *dsrE*, *dsrF*, and *dsrH*). In order to know whether the putative MeHg producers and degraders are potential SRB, the eight genes carried by our focal MAGs were identified by their corresponding KEGG Orthology numbers (Table S11), and the *dsrD* gene was identified according to the methods of Anantharaman et al. (47). Similarly, the identification of potential IRB was based on the presence of *omcF* and *omcS* (Table S11) (73). In addition, several of the other important metabolic pathways (including the nitrogen metabolism and metal tolerance pathways) of our focal MAGs were predicted by the KEGG or eggNOG Orthology numbers for related genes, which are also shown in Table S11.

**Taxonomic classification, abundance calculation, and phylogenetic tree construction.** A total of 70 MAGs that contained *hgcAB* and 192 MAGs that contained *merB* were identified. After dereplication using dRep (version 2.5.4) with the default parameters (74), 46 and 93 nonredundant MAGs containing *hgcAB* and *merB* were obtained, respectively (Tables S2 and S4). The taxonomic assignments of these two kinds of MAGs were inferred from the phylogenetic trees constructed with the reference genomes using GTDB-Tk (version 1.0.2) with the default parameters (66).

The relative abundances of the dereplicated MAGs were calculated as previously described (35, 75). Briefly, the HQ reads from each metagenomic data set were mapped to all of the dereplicated MAGs using BBMap (version 38.44) with the parameters "k = 14, minid = 0.97, and build = 1". The coverage of each MAG was calculated as the average scaffold coverage, which weighted each scaffold by the corresponding length in base pairs. The final relative abundance of each MAG in each sample was calculated as its coverage divided by the total coverage of all of the MAGs in the corresponding metagenomic data set. Furthermore, the occurrence frequency of each MAG was calculated as the number of samples in which that MAG had a relative abundance of greater than zero divided by the total sample number, and the occurrence frequency of each taxon was calculated as the average occurrence frequency of all of the MAGs in that taxon.

The 46 *hgcAB*-containing and 93 *merB*-containing MAGs were separately used to construct phylogenetic trees using PhyloPhlan (version 0.99) (76). The alignment of the HgcAB and MerB sequences were used to separately construct maximum likelihood trees using RAxML (version 8.2.12) (77) with the parameters "-f a -m PROTGAMMALG -p 26565845 -x 12435454 -# 100" (18). The Newick files with the consensus tree were uploaded to the Interactive Tree of Life (iTOL) online interface for visualization and formatting (78).

**Construction of trees to infer the HGT events of *hgcAB* and *merB*.** First, the 70 *hgcAB*-containing MAGs (containing replications) that were recovered in this study and 84 other *hgcAB*-containing genomes that were downloaded from the NCBI GenBank were used for the construction of a genome-based phylogenetic tree using PhyloPhlAn with the default parameters (76). The genomes affiliated with *Euryarchaeota* were selected to be the root of the tree. Then, the *hgcAB* gene tree was constructed using all of the HgcA and HgcB sequences that corresponded to the genome-based phylogenetic tree. The HgcA and HgcB sequences were individually aligned using MUSCLE (version 3.8.1551), and they were trimmed using TrimAl (version 1.4.rev15) with the parameters "-gt 0.95 -cons 50" (79). The alignments of HgcA and HgcB were then concatenated, and the gene tree was constructed using RAxML with the parameters set as described above. The Newick files of the protein tree and the species tree with the bootstrap consensus tree were uploaded to iTOL for rerooting, visualization, and formatting. The MerB species tree was constructed using 192 *merB*-containing MAGs that were obtained in this study (including replications) and 39 other *merB*-containing genomes that were downloaded from the NCBI GenBank. The methods for the alignment and the phylogenetic tree construction of MerB were the same as those used for HgcAB. Finally, all of the putative HGT events were calculated using the same method as described by Anantharaman et al. (47), in which each mismatching branch between the gene tree and the species tree was calculated as a single HGT event.

**Statistical analyses.** All of the statistical analyses were carried out in the statistical program R. The correlations between the relative abundances of *hgcAB*-containing or *merB*-containing MAGs that are affiliated with individual phyla or classes of *Proteobacteria*, selected environmental factors (i.e., the physicochemical properties listed in Table S1), and the concentrations of MeHg in the sediments of the individual mine sites were assessed via a Pearson correlation analysis (the "rcorr" function in the Hmisc package). After the factors that were significantly ($P < 0.05$) correlated with the MeHg concentration were identified, their abilities to explain the variance of the accumulation of MeHg were further calculated using a multiple linear regression model that was optimized via best subsets selection, which was based on the Akaike information criterion, by using the "regsubsets" function in the Leaps package. The estimates of the relative importance of different factors in explaining the variance of the accumulation of MeHg were furnished by the relative weights using the "relweights" function in the Relaimpo package (80, 81). The individual and interactive impacts of the microbial and environmental factors on the accumulation of MeHg were quantified via a variance decomposition approach using the "varpart" function in the vegan package. All of the statistical analyses were performed on $\log_{10}$-transformed data.

**Data availability.** All of the assembled genomes in this study are available on NCBI through the BioProject accession number PRJNA830440, and the BioSample accession numbers for the individual genomes of the putative MeHg producers and degraders are listed in Table S12. Fig. S1–S13, Tables S11

and S12, and the supplemental material (both methods and results) that support the findings in this research were submitted to Figshare (https://doi.org/10.6084/m9.figshare.21516258).

## SUPPLEMENTAL MATERIAL

Supplemental material is available online only.

**TABLE S1**, XLSX file, 0.01 MB.
**TABLE S2**, XLSX file, 0.01 MB.
**TABLE S3**, XLSX file, 0.01 MB.
**TABLE S4**, XLSX file, 0.02 MB.
**TABLE S5**, XLSX file, 0.02 MB.
**TABLE S6**, XLSX file, 0.01 MB.
**TABLE S7**, XLSX file, 0.01 MB.
**TABLE S8**, XLSX file, 0.01 MB.
**TABLE S9**, XLSX file, 0.02 MB.
**TABLE S10**, XLSX file, 0.01 MB.

## ACKNOWLEDGMENTS

We thank A.J.M. Baker (Universities of Melbourne and Queensland, Australia, and Sheffield, UK) for his help with the improvement of the paper. This work was supported financially by the National Natural Science Foundation of China (No. 41830318, 41622106, 42177009, 42077117 and 41907211), the Key-Area Research and Development Program of Guangdong Province (No. 2019B110207001), and the Natural Science Foundation of Guangdong Province of China (No. 2020A1515010937 and 2020A1515110972).

The authors declare that they have no competing interests.

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
