## [Reviewer comments · mSystems]

Diverse methylmercury (MeHg) producers and degraders inhabit acid mine drainage sediments but only a few taxa correlate with MeHg accumulation

Jin Zheng, Jie-Liang Liang, Pu Jia, Shi-wei Feng, Jing-li Lu, Zhen-hao Luo, Hong-xia Ai, Bin Liao, Wen-Sheng Shu, and Jin-tian Li

Corresponding Author(s): Jin-tian Li, South China Normal University

Review Timeline:

Submission Date:	August 3, 2022
Editorial Decision:	October 4, 2022
Revision Received:	November 8, 2022
Accepted:	November 16, 2022

Editor: Hans Bernstein

Reviewer(s): Disclosure of reviewer identity is with reference to reviewer comments included in decision letter(s). The following individuals involved in review of your submission have agreed to reveal their identity: Giovanni Gallo (Reviewer #1)

Transaction Report:

DOI: <https://doi.org/10.1128/msystems.00736-22>

October 4, 2022

Prof. Jin-tian Li
South China Normal University
No. 55, Zhongshan Avenue West
Guangzhou
China

Re: mSystems00736-22 (Diverse methylmercury (MeHg) producers and degraders inhabit acid mine drainage sediments but only a few taxa correlate with MeHg accumulation)

Dear Prof. Jin-tian Li:

Thank you for submitting your manuscript to mSystems. We have completed our review and I am pleased to inform you that, in principle, we expect to accept it for publication in mSystems. However, acceptance will not be final until you have adequately addressed the reviewer comments.

The reviewers were overall impressed with the amount and quality of work presented here and agree that it will be an important contribution towards our understanding of the diversity and metabolic potential within acid mine drainage environments. Please pay specific attention to the reviewer #2's comment regarding the descriptions/wording of "novel" vs. "putative" MeHg producers or degraders. The reviewer felt (and I agree) that over interpretation and confusion arises as to what "putative" should imply. Please be clear in that this study is based in metagenomics and therefore can only explain the genetic potential instead of actually establishing if specific taxa are producing or degrading MeHg. Please also be rigorous in your responses with respect to reviewer #3's critique of additional clarification with respect to methods and approaches (i.e., change and/or add to the text where appropriate). I would also request that you reduce the number of supplementary files to below 10, as is policy with MSystems.

Preparing Revision Guidelines

Sincerely,

Hans Bernstein

Editor, mSystems

Journals Department
Reviewer comments:

Reviewer #1 (Comments for the Author):

In this manuscript entitled 'Several methylmercury (MeHg) producers and degraders populate acid mine drainage sediments, but only a few taxa are related to MeHg accumulation', Zheng and co-authors describe the microbial populations inhabiting MeHg-rich areas in southern China.

I find this work well done and extensive (very long) but not particularly lacking from my point of view; my only curiosity is whether it is possible to indicate the temperature and pH conditions of the sample analysed. It would also be important to specify whether it is mud, soil, or aqueous

Reviewer #2 (Comments for the Author):

The authors present a very large and valuable dataset containing 139 highly complete metagenome assembled genomes (MAGs). They further analyzed these MAGs in their involvement in MeHg production and degradation. They were able to collect their samples from 20 different acid mine drainage sites. They further looked into how these taxa and other environmental factors correlated with MeHg accumulation at these sites. Furthermore the authors determined what percentage of their taxa acquired MeHg production and degradation genes via horizontal gene transfer much like McDaniel et al. did. They found similar results to McDaniel et al although they propose that horizontal gene transfer may be happening at a much faster rate in acid mine drainage sites.

Overarching comments:

Overall the authors present an easy to follow narrative with data that will add value to the field. Although, as the authors point out in their conclusion, using metagenomics, while valuable, has its limitations. The authors occasionally state that certain taxa are MeHg producers or degraders when they cannot know for sure. They should either reference previous work or make sure to clarify it is descriptions based solely on genetic potential. For the most part the authors use putative to describe the MeHg producers or degraders throughout the paper. This is okay for the most part but there are a few times where describing these MAGs as putative is confusing. For it is not commonly known for many of these MAGs that they are MeHg producers or degraders. For example, the authors describe certain taxa as being both putative and novel. It seems odd that they can be both.

Other comments:

Line 29: This is the first time we see acid mine drainage and you don't explain it as AMD. Later in the abstract and Importance section you use AMD and we don't know what it means-

Line 44: Use AMD but don't know what it is

Line 50: same

Line 58: physicochemical property is smaller text for some reason

Line 148 - 149: If these organisms were not previously known MeHg metabolizing microorganisms how can you be confident they are now? They just have the genetic potential to do so

Line 151-152: Again do you know these are for sure degraders and producers?

Line 341-342: This sentence is confusing. I am not exactly sure what you are saying

Line 342: Novel putative seems odd. How is it both

Line 372: When you say 'our study' do you mean this study?

Line 375-376: Again this sentence is confusing

Line 440-441: This is a repeat statement from lines 434-435 but actually uses two different percentages (62.5 % or 63%)

Line 488-491: This is confusing? How was it not novel and how was it novel?

Line 494-501: This whole section on novel putative MeHg producers taxa is confusing. It also seems odd to have so many higher order novel taxa.

Line 523-526: I am not sure what you mean by this statement.

Line 565-567: It seems odd to have an explanation be that they are all extremely low abundance when all of the MAGs at these

sites are below 14% and are all low abundance.

Line 637-638: You are comparing the total of 2 factors to the total of 3 factors it is not surprising that 3 factors are larger than 2.

Line 661-663: The diversity of organisms containing genes for MeHg production and degradation is larger than thought before. These organisms are still not known to actually produce or degrade MeHg.

Reviewer #3 (Comments for the Author):

The authors examined mercury cycling microbial species from 86 AMD sites using metagenomics. They generated metagenome assembled genomes of species containing genes for HgcAB and merG including many novel species. They used phylogenetics to look for signs of lateral gene transfer and correlated the community composition with abiotic parameters. Overall, this appears to have been an incredible amount of work and it contributes to our understanding of the diversity of these organisms. However, I have several key concerns:

Major concerns.

The lateral gene transfer event work was poorly described and difficult to interpret. This needs to be re-written so that it is easier to 1) understand the methods, results, and potential implications and 2) evaluate the science.

While the large scale sections are well organized, paragraphs are often not well organized making it difficult to follow.

There is significant interpretation in the results section. Sections that have interpretation need to be removed and moved into the discussion.

Minor Comments

Lines 152 -160: This summary of the research results should not be present in the introduction. Please remove.

Line 169: Does this mean the total volume of the sample? Was the depth sampled consistent across sites?

Line 190: This kit calls for an input of up to 500 mg of sediment or soil. Were these extracted 2-10 times each or did you use more material that was indicated in the protocol.

Line 207: Which version of MetaBat was used? each of these methodologies should be cited.

Line 208: What was included in the manual examination?

Line 220: HMMR should be cited.

Line 272 - 279: How did you select the appropriate model for generating the tree?

Line 267: A lot of data is lost in the binning stage, so this is not a great way to calculate the relative abundance. It would be better to calculate it using the coverage divided by the coverage of all contigs or something similar.

Line 275: These trees should be bootstrapped with at least 100 if not 1000 bootstraps and the bootstrap consensus tree should be used. The Figure 2 captions suggests that you did bootstrapping. If so, this should be indicated in the method and the consensus tree should be used. (Line 293 suggests that the best topology tree was used).

Line 290: The L in TrimAl should be lower case. The version of TrimAl should be indicated.

Line 293: See note one line 275 above.

Line 300: Which programs/software were used to do the statistical analyses?

Line 339: phylum should be phyla

Line 339-340: This is interpretation and should be in the discussion.

Line 343 - 346: This sentence is difficult to parse.

Line 365: Was should be were.

Line 428: Did you do any corrections for multiple analyses here? They should be corrected using something like a Bonferroni correction.

Line 450: This is interpretation and should not be in the results.

Line 516: This sentence is difficult to understand.

Line 584 -585: I do not think this assertion is supported.

Line 638 - 639: this statement requires a citation.

Line 641 - 642: This statement does not follow for me. If the generation and the use of MeHg are microbial processes, then why would it affect the accumulation of MeHg. Further, many AMD sites with high iron concentrations have robust, active microbial communities, often with the organic carbon generated by autotrophs .

Section 3.4.: Much of this appears to be interpretation rather than results. You should present the topology of the trees without interpreting what they mean.

Figure 1. The concentrations in section B are confusing. What does the acronym Lb mean? What do you mean in the figure caption when you say that the data illustrated as concentration normalized to lg? Do you mean you log transformed the data (as I think is mentioned in line 313)?

Figure 4: The numbers that indicate the number of lateral gene transfers per group are confusing. Perhaps the total number could be indicated on the left hand tree and have putative LGT events indicated as a shape. As it is, initially it looks like single species have 2+ lateral gene transfer events associated with them. The collapsed tree topology also makes it difficult to interpret the LGT events. A tree showing individual species may be easier to interpret.

In this manuscript entitled 'Several methylmercury (MeHg) producers and degraders populate acid mine drainage sediments, but only a few taxa are related to MeHg accumulation', Zheng and co-authors describe the microbial populations inhabiting MeHg-rich areas in southern China.

I find this work well done and extensive (very long) but not particularly lacking from my point of view; my only curiosity is whether it is possible to indicate the temperature and pH conditions of the sample analysed. It would also be important to specify whether it is mud, soil, or aqueous

Responses to the referee comments

Responses to the Editor's remarks

The reviewers were overall impressed with the amount and quality of work presented here and agree that it will be an important contribution towards our understanding of the diversity and metabolic potential within acid mine drainage environments. Please pay specific attention to the reviewer #2's comment regarding the descriptions/wording of "novel" vs. "putative" MeHg producers or degraders. The reviewer felt (and I agree) that over interpretation and confusion arises as to what "putative" should imply. Please be clear in that this study is based in metagenomics and therefore can only explain the genetic potential instead of actually establishing if specific taxa are producing or degrading MeHg. Please also be rigorous in your responses with respect to reviewer #3's critique of additional clarification with respect to methods and approaches (i.e., change and/or add to the text where appropriate). I would also request that you reduce the number of supplementary files to below 10, as is policy with MSystems.

Response: Thanks very much for handling our manuscript and for your remarks. We found that the reviewers' comments are very constructive in further improving the quality of our manuscript. In the revised manuscript (RM), we have tried to address the reviewers' comments as fully as we could. Firstly, we have carefully revised the descriptions/wording of "novel" vs. "putative" MeHg producers or degraders as per the Reviewer 2's comments. That is, the simultaneous usage of both "putative" and "novel"

in a sentence has been avoided throughout the revised manuscript. Secondly, we have added additional clarification with respect to methods and approaches according to Reviewer 3's comments. Thirdly, as suggested by you, we have checked the number supplementary files of our manuscript and made sure that it is in line with the mSystems's policy. The revised manuscript now contains ten supplementary tables (i.e. Tables S1-S10), with other supplemental tables (i.e. Table S11 and S12), figures (i.e. Figs. S1-S13), methods and results being submitted to the Figshare (<https://doi.org/10.6084/m9.figshare.21516258>). Below, please find our point-by-point responses to the reviewers' comments. We hope you will find that our manuscript is improved considerably and suitable for publication in the journal.

Responses to the Reviewer #1's comments

In this manuscript entitled 'Diverse methylmercury (MeHg) producers and degraders inhabit acid mine drainage sediments but only a few taxa correlate with MeHg accumulation', Zheng and co-authors describe the microbial populations inhabiting MeHg-rich areas in southern China.

I find this work well done and extensive (very long) but not particularly lacking from my point of view; my only curiosity is whether it is possible to indicate the temperature and pH conditions of the sample analysed. It would also be important to specify whether it is mud, soil, or aqueous.

Response: We thank this reviewer for acknowledging the merits of our work and providing specific suggestions to us. In the revised manuscript (RM), the temperature

and pH conditions of the samples were provided in Table S1 in the supplementary files. In addition, we have specified that the AMD sediment samples we collected were mud (RM: Line 508).

Responses to the Reviewer #2's comments

The authors present a very large and valuable dataset containing 139 highly complete metagenome assembled genomes (MAGs). They further analyzed these MAGs in their involvement in MeHg production and degradation. They were able to collect their samples from 20 different acid mine drainage sites. They further looked into how these taxa and other environmental factors correlated with MeHg accumulation at these sites. Furthermore they determined what percentage of their taxa acquired MeHg production and degradation genes via horizontal gene transfer much like McDaniel et al. did. They found similar results to McDaniel et al although they propose that horizontal gene transfer may be happening at a much faster rate in acid mine drainage sites.

Overarching comments:

Overall the authors present an easy to follow narrative with data that will add value to the field. Although, as the authors point out in their conclusion, using metagenomics, while valuable, has its limitations. The authors occasionally state that certain taxa are MeHg producers or degraders when they cannot know for sure. They should either reference previous work or make sure to clarify it is descriptions based

solely on genetic potential. For the most part the authors use putative to describe the MeHg producers or degraders throughout the paper. This is okay for the most part but there are a few times where describing these MAGs as putative is confusing. For it is not commonly known for many of these MAGs that they are MeHg producers or degraders. For example, the authors describe certain taxa as being both putative and novel. It seems odd that they can be both.

Response: We thank this reviewer for acknowledging the merits of our work and providing very constructive suggestions to us. We agree well with the reviewer that we should either reference previous work or clarify that the descriptions of MeHg producers or degraders in this study were based solely on genetic potential. Therefore, in the revised manuscript (RM), we have carefully revised the descriptions/wording of "novel" vs. "putative" MeHg producers or degraders as per the reviewer's comments. That is, the simultaneous usage of both "putative" and "novel" in a sentence has been avoided throughout the RM (e.g. Lines 170-173, 317-320, and 487). Below, please find our point-by-point responses to your comments. We hope you will find that our manuscript is improved considerably and suitable for publication in the journal.

Other comments:

Line 29: This is the first time we see acid mine drainage and you don't explain it as AMD. Later in the abstract and Importance section you use AMD and we don't know what it means-

Response: We have changed "acid mine drainage" into "acid mine drainage (AMD)"

in Line 28 in the RM.

Line 44: Use AMD but don't know what it is

Response: AMD is the abbreviation of “acid mine drainage”. For clarification, we have changed “acid mine drainage” into “acid mine drainage (AMD)” in Line 28 in the RM.

Line 50: same

Response: For clarification, we have changed “acid mine drainage” into “acid mine drainage (AMD)” in Line 28 in the RM.

Line 58: physicochemical property is smaller text for some reason

Response: The font size was modified in the RM (Line 58).

Line 148 - 149: If these organisms were not previously known MeHg metabolizing microorganisms how can you be confident they are now? They just have the genetic potential to do so

Response: We agree well with the reviewer that the descriptions of MeHg producers or degraders in this study were based solely on genetic potential. In the RM, “MeHg-metabolizing microorganisms” has been replaced with “putative MeHg-metabolizing microorganisms” (Lines 143).

Line 151-152: Again do you know these are for sure degraders and producers?

Response: In the RM, we deleted this sentence, as the Reviewer #3 believed that this summary of the research results should not be present in the Introduction section.

Line 341-342: This sentence is confusing. I am not exactly sure what you are saying

Response: In the RM, we have rewritten this sentence as “Most of the putative MeHg producers (42/46) were ubiquitous in AMD sediments, as they individually could be observed in more than 50% of the samples (Fig. 2a).” (Lines 170-171).

Line 342: Novel putative seems odd. How is it both

Response: In the RM, we have deleted the word “novel” (Line 172).

Line 372: When you say 'our study' do you mean this study?

Response: Yes. For better understanding, “our study” has been replaced with “this study” in the RM (Line 202).

Line 375-376: Again this sentence is confusing

Response: We have revised the sentence as “Frequencies of occurrence of the *merB*-containing MAGs in all the samples ranged from 58% to 100%” in the RM (Lines 205-206).

Line 440-441: This is a repeat statement from lines 434-435 but actually uses two

different percentages (62.5% or 63%)

Response: Thanks for the comment. We want to note that these two sentences were not repeat statements, as the two different percentages were generated from two different statistical analyses. More specifically, the figure (62.5%) showed in Lines 265-266 in the original manuscript was obtained by multiple regression analysis, which was performed using the `relweigts` function in the `Relaimpo` package; while, the figure showed in Lines 271-272 (63%) in the original manuscript was achieved by variance decomposition.

Line 488-491: This is confusing? How was it not novel and how was it novel?

Response: To avoid confusion, we have rewritten the sentences as “Compared with the taxonomic affiliation based on the NCBI taxonomy, all of the Hg-methylating MAGs in this study were belonged to the phyla with previously known MeHg producers” (RM: Lines 320-322).

Line 494-501: This whole section on novel putative MeHg producers taxa is confusing.

It also seems odd to have so many higher order novel taxa.

Response: Thanks for the comment. To avoid confusion, we have rewritten this section as in Lines 324-332 in the RM. As to the recovery of so many higher order novel taxa, a possible reason is that the environmental conditions of the studied AMD sediments were dramatically different from those of other habitats investigated by the comparable previous studies (Capo et al., 2020; McDaniel et al., 2020; Lin et al.,

2021). In a wider context, many novel microbial taxa (MAGs) have been found in AMD systems, and some of them always contain a high percentage of genes without known biological functions (Shu & Huang, 2022).

Line 523-526: I am not sure what you mean by this statement.

Response: In the RM, we have rewritten this sentence as “Note also that average occurrence frequencies of the putative MeHg producers and degraders from these taxa were greater than 86% in the samples (Table S8), indicating that they were widespread in the AMD sediments.” (Lines 523-526). In addition, it has been specified that “occurrence frequency of each MAG was calculated as the number of samples with relative abundance of that MAG greater than zero divided by the total sample number, and the occurrence frequency of each taxon was calculated as the average occurrence frequency of all MAGs in that taxon.” (RM: Lines 611-615).

Line 565-567: It seems odd to have a explanation be that they are all extremely low abundance when all of the MAGs at these sites are below 14% and are all low abundance.

Response: In the RM, we have rewritten this sentence (Lines 394-397).

Line 637-638: You are comparing the total of 2 factors to the total of 3 factors it is not surprising that 3 factors are larger than 2.

Response: In the RM, we deleted this content.

Line 661-663: The diversity of organisms containing genes for MeHg production and degradation is larger than thought before. These organisms are still not known to actually produce or degrade MeHg.

Response: Agree with this reviewer. In the RM, “MeHg-metabolizing microorganisms” has been changed into “putative MeHg-metabolizing microorganisms”(Line 487).

Responses to the Reviewer #3’s comments

The authors examined mercury cycling microbial species from 86 AMD sites using metagenomics. They generated metagenome assembled genomes of species containing genes for HgcAB and merG including many novel species. They used phylogenetics to look for signs of lateral gene transfer and correlated the community composition with abiotic parameters. Overall, this appears to have been an incredible amount of work and it contributes to our understanding of the diversity of these organisms. However, I have several key concerns:

Major concerns.

The lateral gene transfer event work was poorly described and difficult to interpret. This needs to be re-written so that it is easier to 1) understand the methods, results, and potential implications and 2) evaluate the science.

While the large scale sections are well organized, paragraphs are often not well organized making it difficult to follow.

There is significant interpretation in the results section. Sections that have interpretation need to be removed and moved into the discussion.

Response: We thank this reviewer for acknowledging the merits of our work and providing very constructive comments to us. We have revised our manuscript following your comments. First of all, the methods, results and discussions regarding the lateral gene transfer event work have been largely rewritten (revised manuscript, RM: Lines 640-643, 233-235, 237-239, 243-251 and 413-414). Secondly, some paragraphs have been re-organized for better understanding (e.g. RM: Lines 339-344, Lines 483-499). Thirdly, the interpretations in the results section have been moved to the discussion section as suggested (e.g. RM: Lines 483-486). Below, please find our point-by-point responses to your comments. We hope you will find that our manuscript is improved considerably.

Minor Comments

Lines 152 -160: This summary of the research results should not be present in the introduction. Please remove.

Response: In the RM, we have removed the summary of the research results from the introduction section.

Line 169: Does this mean the total volume of the sample? Was the depth sampled

consistent across sites?

Response: In the RM, we have added more information on the sample collection. The total volume of the sample was about 250 ml (RM: Line 508). Although the depth of AMD varied from site to site, our samples were collected at a depth of 5 ± 1 cm below the water-sediment interface (RM: Lines 509).

Line 190: This kit calls for an input of up to 500 mg of sediment or soil. Were these extracted 2-10 times each or did you use more material that was indicated in the protocol.

Response: These were extracted 2-10 times. The total DNA of AMD sediments is more difficult to extract than those of sediments of other sources (e.g. freshwater). In order to obtain enough DNA from individual samples for metagenomic sequencing, multiple extractions were performed for each sample using the standard protocol, then the DNA from the multiple extractions were combined for that sample. In the RM, we have added a sentence to describe the details (Lines 533-535).

Line 207: Which version of MetaBat was used? each of these methodologies should be cited.

Response: In the RM, we have added the version number (v.2.12.1) for this software and cited a reference for each of these methodologies as suggested (Lines 550).

Line 208: What was included in the manual examination?

Response: In the RM, it has been noted that the manual examination mainly included removing of scaffolds with conflicting phylum-level taxonomy (Lines 551-552).

Line 220: HMMER should be cited.

Response: Done as suggested (RM: Lines 562, 563).

Line 272 - 279: How did you select the appropriate model for generating the tree?

Response: We referred to the method of Hg methylation of MAG conformational trees as described by Jones et al. (2019), using the LG model of amino acid substitution for generating the tree (RM: Line 620).

Line 267: A lot of data is lost in the binning stage, so this is not a great way to calculate the relative abundance. It would be better to calculate it using the coverage divided by the coverage of all contigs or something similar.

Response: Thanks for the comment. We calculated the relative abundances of these MAGs according to the popular method (Emerson et al., 2018; Tan et al., 2019), in which the coverage of one MAG of interest was divided by the coverage of all MAGs recovered from the metagenomic dataset. Note that all the MAGs recovered from the metagenomic dataset rather than only the high-quality MAGs were considered when we did the calculation. As such, most data were used for the calculation of the relative abundance. For instance, the high-quality metagenomic reads of the samples collected from the SKS site mapped to all the MAGs and the high-quality metagenomic reads

mapped to all the scaffolds were very similar: the mapped reads of all MAGs was 10,193,386, while the mapped reads of all scaffolds was 11,992,300. So, we feel that the method we used to calculate the relative abundances of individual focal MAGs is reasonable.

Line 275: These trees should be bootstrapped with at least 100 if not 1000 bootstraps and the bootstrap consensus tree should be used. The Figure 2 captions suggests that you did bootstrapping. If so, this should be indicated in the method and the consensus tree should be used. (Line 293 suggests that the best topology tree was used).

Response: Thanks for the suggestion. We actually used the consensus tree with 100 bootstraps. Therefore, in the RM, “the best topology tree” has been replaced with “the consensus tree” (Lines 621, 635). Additionally, the bootstraps values have been indicated in the phylogenetic trees in Figures 2-5.

Line 290: The L in TrimAl should be lower case. The version of TrimAl should be indicated.

Response: Thank you for the reminders. In the RM, “TrimAL” has been replaced with “TrimAl”, and the version number (v1.4.rev15) of this software has been indicated (Line 632).

Line 293: See note one line 275 above.

Response: Thank you for your comments. In the RM, “the best topology tree” has

been replaced with “the bootstrap consensus tree” (Line 635).

Line 300: Which programs/software were used to do the statistical analyses?

Response: In the RM, we have added the detailed information on the statistical analyses. All statistical analyses were carried out through various packages in the statistical program R. Pearson correlation analysis were performed using “rcorr” function in the Hmisc package. The function “regsubsets” in the Leaps package was used to screen the best model for multiple regression analysis. The relative importance of each variable was predicted using “relweighth” function in the Relaimpo package. Variance decomposition was analyzed using “varpart” function in the vegan package (Lines 646, 651, 655-656, 657-658, 660-661).

Line 339: phylum should be phyla

Response: Done as suggested (RM: Line 168).

Line 339-340: This is interpretation and should be in the discussion.

Response: Done as suggested (RM: Lines 317-320).

Line 343 - 346: This sentence is difficult to parse.

Response: In the RM, we have rewritten this sentence as “The relative abundances of three MAGs (i.e. BPO.bin3, FAK.bin3, and FAK.bin5) were > 0.30% respectively, higher than the average relative abundances of *hgcAB*-containing MAGs across all

samples (0.06%).” (Lines 173-176).

Line 365: Was should be were.

Response: Done as suggested (RM: Line 195).

Line 428: Did you do any corrections for multiple analyses here? They should be corrected using something like a Bonferroni correction.

Response: Thanks for the comment. In this study, we used multiple regression analysis instead of multiple comparison analysis. R^2 was automatically adjusted in multiple regression analysis (Table S6).

Line 450: This is interpretation and should not be in the results.

Response: Done as suggested (RM: Lines 448-450).

Line 516: This sentence is difficult to understand.

Response: In the RM, we have rewritten this sentence as “In this study, the putative MeHg degraders belonged to eight phyla, among which six have not been reported to display MeHg-demethylation ability previously.” (RM: Lines 346-347).

Line 584 -585: I do not think this assertion is supported.

Response: In the RM, we have rewritten this sentence as “These findings indicate that the evolutionary history of *hgcAB* genes in *Deltaproteobacteria* could be very

complex.” (RM: Lines 413-414).

Line 638 - 639: this statement requires a citation.

Response: In the RM, a citation has been added (RM: Lines 466).

Line 641 - 642: This statement does not follow for me. If the generation and the use of MeHg are microbial processes, then why would it affect the accumulation of MeHg. Further, many AMD sites with high iron concentrations have robust, active microbial communities, often with the organic carbon generated by autotrophs.

Response: Thanks for the comment. Among the 46 *hgcAB*-containing MAGs and 93 *merB*-containing MAGs recovered in our study, only 2 MAGs could be considered as potential autotrophs as they harbored the complete CBB or rTCA pathway. That is, the majority of the putative MeHg producers or degraders (137/139) could be considered as heterotrophs, which rely on the organic carbon in AMD sediments for growth and metabolism. In this context, we feel that it is reasonable to assume that the insufficient carbon sources in AMD sediments likely limit the activity of MeHg-metabolizing microorganisms, which in turn can affect the MeHg accumulation in AMD sediments.

Section 3.4. Much of this appears to be interpretation rather than results. You should present the topology of the trees without interpreting what they mean.

Response: Thanks for the comment. In the RM, we have largely written this section (Lines 233-235, 237-239 and 241-251).

Figure 1. The concentrations in section B are confusing. What does the acronym Lb mean? What do you mean in the figure caption when you say that the data illustrated as concentration normalized to lg? Do you mean you log transformed the data (as I think is mentioned in line 313)?

Response: To avoid confusion, the acronym “Lg” in the Figure 1 has been replaced with “Log₁₀” (RM: Figure 1). Meanwhile, it has been specified in the figure caption that “The log₁₀-transformed concentration data are shown.” (RM: Lines 939-940).

Figure 4: The numbers that indicate the number of lateral gene transfers per group are confusing. Perhaps the total number could be indicated on the left hand tree and have putative LGT events indicated as a shape. As it is, initially it looks like single species have 2+ lateral gene transfer events associated with them. The collapsed tree topology also makes it difficult to interpret the LGT events. A tree showing individual species may be easier to interpret.

Response: Thanks for the comment. In the RM, we have redrawn Figures 4 and 5 according to your constructive suggestions. In the two new figures, the total number of independent HGT events in specific phyla/classes are indicated on the left tree, and the putative HGT events are indicated with triangles on the right tree. We preferred to use the collapsed trees to illustrate the mismatching branching pattern of the genomic-based phylogenetic tree and the protein tree, considering that a tree showing individual species is difficult to compare the two trees (Lines 981-984, 995-998).

References

- Emerson JB, Roux S, Brum JR, Bolduc B, Woodcroft BJ, Jang HB, Singleton CM, Solden LM, Naas A.E, Boyd JA, Hodgkins SB, Wilson RM, Trubl G, Li, C.S., Frolking S, Pope PB, Wrighton KC, Crill PM, Chanton JP, Saleska SR, Tyson GW, Rich VI, Sullivan MB. 2018. Host-linked soil viral ecology along a permafrost thaw gradient. *Nat Microbiol* 3:870-880.
- Jones DS, Walker GM, Johnson NW, Mitchell CP, Wasik JKC, Bailey JV. 2019. Molecular evidence for novel mercury methylating microorganisms in sulfate-impacted lakes. *ISME J* 13:1659-1675.
- Shu WS, & Huang LN. 2022. Microbial diversity in extreme environments. *Nat Rev Microbiol* 20:219-235.
- Tan S, Liu J, Fang Y, Hedlund BP, Lian ZH, Huang LY, Li JT, Huang LN, Li WJ, Jiang HC, Dong HL, Shu WS. 2019. Insights into ecological role of a new deltaproteobacterial order *Candidatus Acidulodesulfobacterales* by metagenomics and metatranscriptomics. *ISME J* 13:2044-2057.

November 16, 2022

Prof. Jin-tian Li
South China Normal University
No. 55, Zhongshan Avenue West
Guangzhou
China

Re: mSystems00736-22R1 (Diverse methylmercury (MeHg) producers and degraders inhabit acid mine drainage sediments but only a few taxa correlate with MeHg accumulation)

Dear Prof. Jin-tian Li:

I was not able to obtain a second round from reviewer #3. However, my assessment is that the authors responded robustly to all of the reviewer comments by making changes to the manuscript where needed. I cannot see an instance where the authors argued against one of the reviewer's suggestions and note that the first review was also quite positive. I agree with the initial assessment that this paper represents an important contribution towards our understanding of the diversity and metabolic potential within acid mine drainage environments. The metagenomics-based methodology is sufficiently advanced and the interpretations are supported by the data.

Your manuscript has been accepted, and I am forwarding it to the ASM Journals Department for publication. For your reference, ASM Journals' address is given below. Before it can be scheduled for publication, your manuscript will be checked by the mSystems production staff to make sure that all elements meet the technical requirements for publication. They will contact you if anything needs to be revised before copyediting and production can begin. Otherwise, you will be notified when your proofs are ready to be viewed.

Publication Fees:

If you would like to submit a potential Featured Image, please email a file and a short legend to mSystems@asmusa.org. Please note that we can only consider images that (i) the authors created or own and (ii) have not been previously published. By submitting, you agree that the image can be used under the same terms as the published article. File requirements: square dimensions (4" x 4"), 300 dpi resolution, RGB colorspace, TIF file format.

We recognize that the video files can become quite large, and so to avoid quality loss ASM suggests sending the video file via

<https://www.wetransfer.com/>. When you have a final version of the video and the still ready to share, please send it to mSystems staff at mSystems@asmusa.org.

Sincerely,

Hans Bernstein
Editor, mSystems

Journals Department
Table S10: Accept
Table S2: Accept
Table S9: Accept
Table S4: Accept
Table S5: Accept
Table S7: Accept
Table S6: Accept
Table S1: Accept
Table S3: Accept
Table S8: Accept